# Characterization of aerosol number size distributions and their effect on cloud properties at Syowa Station, Antarctica

Keiichiro Hara[1], Chiharu Nishita-Hara[2], Kazuo Osada[3], Masanori Yabuki[4], and Takashi Yamanouchi[5]

[1] Department of Earth System Science, Faculty of Science, Fukuoka University, Fukuoka, 814-0180, Japan
[2] Fukuoka Institute for Atmospheric Environment and Health, Fukuoka University, Fukuoka, 814-0180, Japan
[3] Graduate School of Environmental Studies, Nagoya University, Nagoya, 464-8601, Japan
[4] Research Institute for Sustainable Humanosphere, Kyoto University, Kyoto, 611-0011, Japan
[5] National Institute of Polar Research, Tokyo, 190-0014, Japan

*Correspondence to*: Keiichiro Hara (harakei@fukuoka-u.ac.jp)

**Abstract.** We took aerosol measurements at Syowa Station, Antarctica to characterize the aerosol number–size distribution and other aerosol physicochemical properties in 2004–2006. Four modal structures (i.e., mono-, bi-, tri-, and quad-modal) were identified in aerosol size distributions during measurements. Particularly, tri-modal and quad-modal structures were associated closely with new particle formation (NPF). To elucidate where NPF proceeds in the Antarctic, we compared the aerosol size distributions and modal structures to air mass origins computed using backward trajectory analysis. Results of this comparison imply that aerosol size distributions involved with fresh NPF (quad-modal distributions) were observed in coastal and continental free troposphere (FT; 12% of days) areas and marine and coastal boundary layers (1%) during September–October and March, and in coastal and continental FT (3%) areas and marine and coastal boundary layers (8%) during December–February. Photochemical gaseous products, coupled with ultraviolet (UV) radiation, play an important role in NPF, even in the Antarctic troposphere. With the existence of the ozone hole in the Antarctic stratosphere, more UV radiation can enhance atmospheric chemistry, even near the surface in the Antarctic. However, linkage among tropospheric aerosols in the Antarctic, ozone hole, and UV enhancement is unknown. Results demonstrated that NPF started in the Antarctic FT already at the end of August – early September by UV enhancement resulting from the ozone hole. Then, aerosol particles supplied from NPF during periods when the ozone hole appeared to grow gradually by vapor condensation, suggesting modification of aerosol properties such as number concentrations and size distributions in the Antarctic troposphere during summer. Here, we assess the hypothesis that UV enhancement in the upper troposphere by the Antarctic ozone hole modifies the aerosol population, aerosol size distribution, cloud condensation nuclei capabilities, and cloud properties in Antarctic regions during summer.

## 1 Introduction

The Antarctic is isolated from human activities occurring in the mid-latitudes. In spite of the slight amount of human activity in the Antarctic, such as research activities at each station and tourism mostly in the Antarctic Peninsula during summer, the source strength of anthropogenic species (e.g., black carbon from combustion processes) is negligible in the Antarctic circle at the moment (e.g., Weller et al., 2013; Hara et al., 2019). Consequently, aerosol measurements in the Antarctic have been taken to ascertain aerosol physicochemical properties, atmospheric chemistry, and their effects on climate change on Earth's background conditions (i.e., cleanest and pristine conditions).

Despite the cleanest conditions prevailing in the Antarctic, concentrations of condensation nuclei (CN) show clear seasonal variations with maximum in summer and minimum in winter (Gras et al., 1993; Hara et al., 2011a; Weller et al., 2011). This seasonal variation relates to supply and deposition processes. In addition to primary aerosol emissions such as sea-salt aerosols from sea-surface and sea-ice regions (e.g., Hara et al., 2020), atmospheric aerosol formation (i.e. new particle formation, NPF)

is important to supply atmospheric aerosols (Kulmala et al., 2013; Kerminen et al., 2018) and to affect climate through indirect processes (Asmi et al., 2010; Dall'Osto et al., 2017). Measurements of aerosol size distributions were taken continuously to elucidate and discuss (1) NPF, (2) particle growth, (3) volatility as indirect information of aerosol constituents, and (4) hygroscopicity and ability of cloud condensation nuclei (CCN) using a scanning mobility particle sizer (SMPS) in the Antarctic (Koponen et al., 2003; Virkkula et al., 2007; Asmi et al., 2010; Hara et al., 2011b; Kyrö et al., 2013; Järvinen et al., 2013;

Weller et al., 2015; Jokinen et al., 2018; Kim et al., 2019; Jang et al., 2019; Lachlan-Cope et al., 2020) and a similar instrument (Ito, 1993). Seasonal features of aerosol number concentrations are associated with primary emissions of sea-salt aerosols, NPF, emissions of aerosol precursors from oceanic bioactivity, and photochemical processes (e.g., Koponen et al., 2003; Virkkula et al., 2007; Asmi et al., 2010; Hara et al., 2011a, 2011b, 2020; Kyrö et al., 2013; Järvinen et al., 2013; Fiebig et al., 2014; Weller et al., 2015; Humphries et al., 2016; Jang et al., 2019; Frey et al., 2020). Aerosol volatility measurements revealed

high abundance of less-volatile particles because of dominance of sea-salt particles, even in ultrafine mode (smaller than 100 nm diameter) during winter – early spring, whereas volatile particles such as $H_2SO_4$ and organics were found to be dominant during spring–summer (Asmi et al., 2010; Hara et al., 2011b; Weller et al., 2011). In polar regions during winter – early spring, sea-salt particles in ultrafine–coarse modes were released from snow and sea-ice surfaces via wind blowing and sublimation (e.g., Hara et al., 2012, 2017, 2020; Frey et al., 2020). In addition to primary emission of sea-salt aerosols in colder seasons,

NPF was observed along the Antarctic coasts during summer at the boundary layer (Koponen et al., 2003; Virkkula et al., 2007; Asmi et al., 2010; Kyrö et al., 2013; Weller et al., 2015; Jokinen et al., 2018; Kim et al., 2019; Jang et al., 2019; Lachlan-Cope et al., 2020) and in the free troposphere (FT; Hara et al., 2011a; Humphries et al., 2016; Lachlan-Cope et al., 2020). Particularly, NPF events at Neumayer, Aboa, and King Sejong during summer were observed in air masses transported from the marine boundary layer with lower or no sea-ice density (Weller et al., 2015; Jokinen et al., 2018; Kim et al., 2019). In

addition to NPF in the marine boundary layer, NPF and high CN concentrations were identified in the FT and the air masses transported from FT (Hara et al., 2011a; Humphries et al., 2016; Lachlan-Cope et al., 2020). Moreover, NPF occurred around melt-ponds on the Antarctic continent during summer (Kyrö et al., 2013). Although most earlier works have specifically examined summer NPF in the Antarctic (Koponen et al., 2003; Virkkula et al., 2007; Asmi et al., 2010; Kyrö et al., 2013; Weller et al., 2015; Jokinen et al., 2018), some investigations (Hara et al., 2011a; Järvinen et al., 2013; Kim et al., 2019) have

emphasized that NPF occurs as early as September in the Antarctic.

Knowledge and discussion of condensable vapors (i.e., aerosol precursors) is fundamentally important to elucidate microphysical processes such as NPF and growth, and locations at which NPF occurs. Earlier works have emphasized examination of the following condensable vapors (i.e., aerosol precursors) for NPF and particle growth: $H_2SO_4$, $CH_3SO_3H$,

$HIO_3$, $NH_3$, amines, and other organics with low vapor pressure (Yu et al., 2012; Kulmala et al., 2013; Kyrö et al., 2013; Weller et al., 2015; Sipilä et al., 2016; Jen et al., 2016; Jokinen et al., 2018; Shen et al., 2019; Burrell et al., 2019). Gaseous $H_2SO_4$ has been regarded as an important aerosol precursor (e.g., Kulmala et al., 2013; Kerminen et al., 2018). Actually, $H_2SO_4$ in the Antarctic is converted dominantly via photochemical oxidation of dimethyl sulfide (DMS) released from biogenic activity in the ocean, and $SO_2$ derived from DMS oxidation (e.g., Minikin et al., 1998; Weller et al., 2015; Enami et al., 2017; Jang et al.,

2019). Indeed, high DMS concentrations in the Antarctic were found in the sea-ice margin and marine boundary layer during summer (Koga et al., 2014). Concentrations of $H_2SO_4$ and $CH_3SO_3H$ showed diurnal change with a maximum in daytime at Palmer Station (Jefferson et al., 1998a, 1998b). Additionally, NPF often occurred in daytime at Neumayer and Aboa during summer (Asmi et al., 2010; Weller et al., 2015; Jokinen et al., 2018). Measurements of aerosol hygroscopicity at Aboa indicated an important role of organic vapors for the growth of freshly nucleated particles (Asmi et al., 2010). Recently, simultaneous

measurements of aerosol precursors and aerosol size distributions exhibited that $H_2SO_4$ and $NH_3$ made important contributions to NPF and growth at Aboa in air masses from the marine boundary layer (Jokinen et al., 2018). Condensable vapors other

than $H_2SO_4$ and $NH_3$, for example iodine compounds such as $HIO_3$ and organics, are associated with atmospheric and snowpack chemistry (e.g., Saiz-Lopez et al., 2008; Atkinson et al., 2012; Roscoe et al., 2015; Sipilä et al., 2016; Hara et al., 2020), and oceanic bioactivity (Dall'Osto et al., 2019). Contributions of the condensable vapors to NPF and growth have been discussed, but this topic has remained unclear for the Antarctic.

Some condensable vapors for NPF and growth are formed through photochemical reactions with atmospheric oxidants such as OH, $O_3$, and BrO (Read et al., 2008). They relate to UV radiation (Matsumi et al., 2002; Matsumi and Kawasaki, 2003). Appearance of the ozone hole in the Antarctic stratosphere during September–November (Hoppel et al., 2005) is expected to enhance UV radiation and atmospheric oxidation potential in the troposphere (Jones et al., 2003). Nevertheless, effects on aerosols in the Antarctic troposphere have not been elucidated sufficiently. Because of low aerosol number concentrations, direct effects of aerosol radiative forcing are negligible in the Antarctic (Bodhaine, 1995). After aerosol activation to cloud condensation nuclei (CCN), indirect effects might affect the atmospheric radiation budget and climate (Bromwich et al., 2012). With the present study, we are striving to understand the occurrence of NPF in the Antarctic, linkage among NPF, the ozone hole, and cloud properties in the Antarctic coast (around Syowa Station).

## 2 Experiments and Analysis

### 2.1 Aerosol measurements at Syowa Station, Antarctica

Aerosol measurements were taken during the 45th–47th Japanese Antarctic Research Expedition (2004–2006) at Syowa Station, Antarctica (69.0 °S, 39.0 °E), located on East Ongul Island in Lützow Holm Bay. Size distributions of ultrafine particles (diameter ($D_p$): 5–168 nm) were measured using SMPS (3936-N-25; TSI Inc.) during February 2004 – December 2006. A condensation particle counter (CPC) was used (3025A; TSI Inc.) to take SMPS measurements. Each scan for SMPS measurements took 5 min. CN concentrations ($D_p$ >10 nm) was monitored using CPC 3010 (TSI Inc.) and was recorded every minute. In addition, an optical particle counter (OPC: KC22B; Rion Co. Ltd.) was used for measurements of size distributions of aerosols with size of $D_p$ >0.08, >0.1, >0.2, 0.3, and >0.5 µm during January 2005 – December 2006. In OPC measurements, number concentrations were recorded every minute. SMPS, CPC, and OPC were operated at room temperature of ca. 20 °C in the clean air observatory located at the windward side ca. 400 m distant from the main area of the station, where a diesel generator was operated. When winds came from the main area, local contamination might have occurred. Before analysis and discussion, locally contaminated data were screened using condensation nuclei concentrations (measured as aerosol monitoring at Syowa Station) and wind data (provided by the Japanese Meteorological Agency). Data screening procedures and criteria were identical to those described in earlier reports (Hara et al., 2011b, 2019). After screening of locally contaminated data, daily mean aerosol number concentrations and size distributions were calculated for data analysis. Because of lower aerosol number concentrations during winter through early spring, data quality of hourly mean aerosol size distributions was insufficient to identify NPF and to analyze growth rate after NPF through the year. Consequently, daily mean aerosol size distribution was analyzed and discussed in this study. Details of the respective NPF events during the periods with higher aerosol concentrations will be discussed elsewhere. Aerosol sampling for chemical analysis was made using a two-stage mid-volume impactor and a back-up filter. Water soluble aerosol constituents such as $Na^+$, $SO_4^{2-}$, and $CH_3SO_3^-$ were determined using ion chromatography. Details of aerosol sampling and chemical analysis were presented in reports of our earlier work (Hara et al., 2004, 2018).

**2.2 Data analysis**

**2.2.1 Estimation of coagulation sink and condensation sink**

Nano-size aerosol particles are removed rapidly through coagulation. To elucidate the speed of removal by coagulation, coagulation sink (*Coag.S*) was calculated using the following equation (Kulmala et al., 2001).

$$\text{Coag.S} = \sum_j K_{ij} N_j \quad (1)$$


Therein, $K_{ij}$ and $N_j$ respectively represent the coagulation coefficient in the transitional regime and the number concentrations of the size bin of $j$.

Condensation sink (*Cond.S*) was calculated to elucidate the rate of condensable vapor removal by condensation on aerosol

particles using the following equation (Kulmala et al., 2001).

$$Cond.S = 4\pi D \int_0^\infty r\beta_M(r)\ n(r)dr = \sum_i \beta_M r_i N_i. \quad (2)$$

Therein, $r$ and $\beta_M$ respectively denote the particle radius and the transitional correction factor given by the following

equation.

$$\beta_M = \frac{Kn+1}{0.377Kn+1+\frac{4}{3}\alpha^{-1}Kn^2+\frac{4}{3}\alpha^{-1}Kn} \quad (3).$$

In that equation, $Kn$ and $\alpha$ respectively represent the Knudsen number and the sticking coefficient. The value of $\alpha$ is

assumed to be unity. Also, $Kn$ is expressed as

$$Kn = \frac{\lambda}{r} \quad (4).$$

In equation (4), $\lambda$ denotes the mean free path. Details of calculation procedures were presented in an earlier report (Kulmala

et al., 2001).

**2.2.2 Estimation of the nucleation rate of aerosol particles**

The nucleation rate of aerosol particles ($D_p$ = 5 nm: $J_5$) was estimated as elucidating NPF and particle growth in the Antarctic troposphere. Number concentrations ($N$) of nano-size aerosol particles in each size bin depend on (1) growth from the smaller size by vapor condensation, (2) coagulation loss, and (3) growth to larger size by condensation. To calculate $J_5$, three size bins

based on particles were used: $D_p$ 1–5 nm, 5–10 nm, and 10–20 nm (Fig. 1). The change of the number concentrations for each size bin can be given by the following equation using *Coag.S* and the condensation growth rate (*Cond*) in accordance with Dal Maso et al. (2002).

$$\frac{dN_{1-5}}{dt} = J_1 - Coag.S_{1-5}N_{1-5} - Cond_{1-5}N_{1-5} \quad (5)$$


$$\frac{dN_{5-10}}{dt} = J_5 - Coag.S_{5-10}N_{5-10} - Cond_{5-10}N_{5-10} \quad (6)$$

$$\frac{dN_{10-20}}{dt} = J_{10} - Coag.S_{10-20}N_{10-20} - Cond_{10-20}N_{10-20} \quad (7)$$

In this procedure, the flux of condensation growth to next size bin ($Cond_iN_i$) in $i$ bin corresponds to $J_{i+1}$. From equations (5)–(7), $J_5$ give the following equations, assuming that $Cond_{10-20}N_{10-20}$ was negligible to estimate $J_5$.

$$J_5 = \frac{(Coag.S_{5-10} + Cond_{5-10})}{Cond_{5-10}}\left[\frac{\Delta N_{10-20}}{\Delta t} + Coag.S_{10-20}N_{10-20}\right] \quad (8)$$

In this study, growth rate after NPF was not estimated in respective NPF events, as described above. Here, the growth rate ($0.01-1$ nm h$^{-1}$) and $\Delta N_{10-20} = 0$ was assumed to estimate $J_5$. Therefore, $J_5$ is obtainable from $Coag.S_{5-10}N_{5-10}$, $Cond_{5-10}$, and $Coag.S_{10-20}N_{10-20}$. Details of procedures for estimation of $Cond_{5-10}$ were described by Dal Maso et al. (2002). It is noteworthy that primary emissions of ultrafine aerosol particles can engender false estimation of $J_5$. Sea-salt particles with less volatility were dominant even among ultrafine aerosol particles with sizes of $D_p > 10$ nm under storm conditions that prevailed during winter–spring at Syowa Station (Hara et al., 2011b). Indeed, a high value of $J_5$ was identified occasionally in the high contribution of sea-salt particles during winter.

### 2.2.3 Lognormal fitting of aerosol size distributions

Multi-modal fitting analysis of aerosol size distributions is used commonly to understand and discuss microphysical processes in the atmosphere. In this study, daily mean aerosol size distributions were approximated by a lognormal function, which is given by the following equation.

$$\frac{dN}{dlogD_p} = \sum_{i=1}^{n}\frac{N_i}{\sqrt{2\pi}log\sigma_i}exp\left[-\frac{(logD_p - logD_{p,i})}{2log^2\sigma_i}\right] \quad (9)$$

In equation (9), $D_p$, $n$, $D_{p,i}$, $\sigma_i$, and $N_i$ respectively denote the particle diameter, mode number ($n = 1$–4), modal size in mode i (i.e., mean diameter of the mode in aerosol number size distributions), modal standard deviation in mode $i$, and the aerosol number concentrations in mode $i$. Lognormal fitting was performed using the nls.lm function of minpack.lm library of R (R interface to the Levenberg–Marquardt nonlinear least-squares algorithm found in MINPACK). To avoid unrealistic lognormal fitting, we set the following restrictions: (1) $N_i > 1\%$ of total particle concentrations, (2) $1.2 \leq \sigma_i \leq 2.2$, and (3) $1.3\ D_{p,i} < D_{p,i+1}$. The daily mean aerosol size distributions ($D_p = 5$–168 nm: SMPS only) were approximated by lognormal functions in 2004. The size distributions ($D_p = 5$–300 nm: SMPS + OPC) were analyzed using lognormal fitting in 2005–2006. For this study, we defined each mode based on the particle size range as follows: fresh nucleation mode ($D_p < 10$ nm), aged nucleation mode ($D_p = 10$–25 nm), first Aitken mode ($D_p = 25$–50 nm), second Aitken mode ($D_p = 50$–100 nm), and accumulation mode ($D_p > 100$ nm).

### 2.3 Backward trajectory analysis

The 120-hr (5-day) backward trajectory was computed using the NOAA-HYSPLIT model (https://ready.arl.noaa.gov/HYSPLIT.php) with the NCEP meteorological dataset (reanalysis) in model vertical velocity mode. The initial point was at 500 m above ground level over Syowa Station, Antarctica because the trajectory at lower altitudes has high uncertainty as a result of topographic effects and turbulence. According to tethered balloon measurements at Syowa Station (Hara et al., 2011a), the top of the boundary layer over Syowa was identified at 1400 m above ground level and lower (annual mean, 840 m). For this study, we use the following criteria to divide each air mass origin: marine, <66 °S; coastal, 66–75°S; Antarctic-continental, >75 °S; boundary layer (BL), <1500 m; and FT, >1500 m. Then, the time passing in each area such as marine BL (MBL), coastal BL, continental BL, continental FT, coastal FT, and marine FT (MFT) was counted for

each backward trajectory. Areas with air masses staying for the longest times in the 5-day backward trajectory were classified into their respective air mass origins.

## 3 Results and Discussion

### 3.1 Aerosol size distributions

Figure 2 presents examples of number–size distributions of aerosol particles observed at Syowa Station. Our measurements show size distributions of ultrafine aerosol particles with mono-, bi-, tri-, and quad-modal structures. In earlier studies (Järvinen et al., 2013; Weller et al., 2015), multi-modal aerosol size distributions with mono-modal, bi-modal, and tri-modal structures were identified in the Antarctic. Although quad-modal structures were observed clearly for this study, they were not described in reports of earlier works.

To characterize the aerosol size distributions, we compare the modal size in each mode (Fig. 3). In mono-modal structure, the modal size ranged mostly in 40–105 nm. In the bi-modal structure, the modal sizes in first and second modes were in the ranges of 20–40 nm and 60–135 nm. In the tri-modal structure, the modal sizes in first–third modes were, respectively, in the ranges of 8–20, 20–63, and 65–135 nm. In the quad-modal structure, the modal sizes in first–fourth modes were, respectively, in the ranges of 7–13, 14–30, 30–65, and 70–140 nm. The mono-modal structure was observed often under storm and strong wind conditions with blowing snow during winter – early spring (Hara et al., 2011b, 2020). In these conditions, sea-salt particles in ultrafine–coarse modes were released from snow and sea-ice surface in polar regions (Hara et al., 2011b, 2014; 2017, 2020; Frey et al., 2020). In the bi-modal structure, modal sizes in the modes of the smallest modal size were greater than those in tri-modal and quad-modal structures, so that the bi-modal structure was well-aged relative to tri-modal and quad-modal structures. In tri-modal and quad-modal structures, modal sizes in the smallest mode appeared mostly for diameters smaller than 20 nm. As demonstrated by Asmi et al. (2010), Kyrö et al. (2013), Järvinen et al. (2013), Weller et al. (2015), Jokinen et al. (2018), and Kim et al. (2019), aerosol particles were grown to a few tens of nanometers after NPF, even in the Antarctic troposphere during summer. Because the smallest mode appeared with diameter smaller than 20 nm, occasionally smaller than 10 nm, in tri-modal and quad-modal structures, aerosol size distributions with tri-modal and quad-modal structures might be associated with NPF and growth by vapor condensation. In bi-, tri-, and quad-modal structures, the modal sizes with the modes with the largest modal sizes had similar diameter larger than 50 nm, which corresponded to critical diameter for CCN activation in the Antarctic (Asmi et al., 2010).

Figure 4 shows the seasonal variation of abundance of modal structures at Syowa Station during our measurements. Mono-modal structure was identified mostly during May–August. Abundance of mono-modal structures was found for 16–60% of days (mean, 37%) during winter. As described above, mono-modal structures during winter – early spring were associated with sea-salt aerosol emissions. Indeed, sea-salt aerosols released from sea-ice areas were dominant during winter – early spring at the Antarctic coasts (e.g., Hara et al., 2012, 2013, 2020; Frey et al., 2020). Although bi-modal structures were observed throughout the year, abundance of bi-modal structures occurred as 23–76% (mean, 56%) in April–September and 9–52% (mean, 26%) in December–March. Particularly, abundance of mono-modal and bi-modal structures were dominant (more than 90%) during May–August. High abundance of tri-modal structures (14–75%, mean 45%) was observed in September–April. Particularly, abundance of tri-modal structures exceeded 50% in January–March. Surprisingly, tri-modal structures were identified even under dusk and polar night conditions during May–August. Modal sizes in the smallest mode of tri-modal structure were greater in winter than in spring–summer (details are presented in a later section). It is noteworthy that the quad-modal structure was found not only in December–February, but also in August–November and March–April. Considering the modal size in the smallest mode of quad-modal distributions, NPF might proceed in August–April in the Antarctic. Indeed,

CN concentrations started to increase in August, with high concentrations in October–February at coastal stations (e.g., Hara et al., 2011a, Weller et al., 2011; Asmi et al., 2013).

## 3.2 Relation between modal structures and air mass history

Annual cycles of air mass origins in each modal structure using 120-hr backward trajectory analysis are shown in Figure 5 for comparison between the modal structure and air mass history. Regarding general features of air mass origins in February 2004 – December 2006, coastal BL was dominant in summer (Fig. 5a), whereas abundance of air masses from continental FT increased during winter. Seasonal cycles of air mass origins in 2004–2006 showed good agreement with results of long-term analysis of air mass origins at Syowa during 2005–2016 (Hara et al., 2019).

Seasonal features of air mass origins in mono-modal structures (Fig. 5b) resembled the general features (Fig. 5a), although the abundance of air masses from coastal FT was slightly higher in August and October. Considering that mono-modal structures corresponded mostly to storm conditions and strong winds during winter–spring (Hara et al., 2010, 2011b, 2020), the appearance of mono-modal structures was associated with primary emissions of sea-salt aerosols from the snow surface on sea-ice by strong winds rather than air mass history (i.e., transport pathway), as presented by Hara et al. (2012, 2013, 2020).

Similarly, seasonal features of air mass origins in bi-modal structure (Fig. 5c) resembled the general features. In general, bi-modal structures were recognized as well-aged distributions by condensation growth, coagulation, and cloud processes. Therefore, the appearance of bi-modal structures might be compared only slightly to air mass origins classified by 120-hr backward trajectory analysis.

Regarding tri-modal structures (Fig. 5d), the abundance of air masses from continental FT and coastal FT increased in spring, compared to general features (Fig. 5a). Similarly to bi-modal structures, the appearance of some tri-modal structures, particularly with larger modal size (e.g., $D_p$>20 nm) in the smallest mode needed aging processes for a longer time. Consequently, seasonal variations of air mass origins in tri-modal structures were similar to general features, although the sum of abundance of air masses from continental FT and coastal FT exceeded 50–60% in August – October. This abundance was slightly higher than that of general features.

Unlike the features in mono-modal, bi-modal, and tri-modal structures, continental FT and coastal FT were the most abundant air mass origins in quad-modal structures during spring and autumn (Fig. 5e). In general, features of air mass origins (Fig. 5a), MBL and coastal BL showed an important contribution during spring and autumn. Nevertheless, quad-modal structures in spring and autumn were identified only in the air masses from continental FT and coastal FT. This feature implies strongly that NPF proceeded in FT during spring and autumn in the Antarctic. In contrast to the high contributions of air masses from continental FT and coastal FT during spring and autumn, quad-modal structures were observed also in air masses from MBL and coastal BL during summer. Therefore, NPF might occur also in MBL and coastal BL during summer, as reported from results of earlier works (Weller et al., 2011; Asmi et al., 2013; Lachlan-Cope et al., 2020).

Based on aspects of location where NPF occur in the Antarctic troposphere, seasonal features of abundance of tri-modal and quad-modal structures and their air mass origins are presented for comparison in Figure 6. The abundance of tri-modal and quad-modal structures reflects the frequency of NPF in the Antarctic troposphere, although the appearance of tri-modal and quad-modal structures does not necessarily mean fresh or local NPF events near Syowa in this study. The abundance of tri-modal and quad-modal structures was less than 10% of days during May–August. In September–January, the abundance of tri-modal and quad-modal structures was 40–48% (mean, 44%). By contrast, abundance reached 60 and 84% in February and

March. Seasonal features of NPF occurrence with highs in spring–summer and minimum in winter were observed at Concordia (Järvinen et al., 2013) and King Sejong (Kim et al., 2019). The monthly occurrence (frequency) of the NPF, however, varied greatly at Syowa, Concordia and King Sejong. Differences of the abundance of NPF occurrence among Syowa, Concordia, and King Sejong might derive from different atmospheric conditions such as the concentrations of aerosols and precursors, and different criteria for identification of NPF-growth events.

Abundance of tri-modal and quad-distributions with air mass origins of coastal FT and continental FT (Fig. 6) ranged in 14–27% of days (mean, 22%) during September–November, in 8–16% (mean, 11%) during December–February, and in 11–32% (mean, 22%) during March–April. Particularly, fresh NPF (quad-modal structures) was identified in FT (12% of days) during September–October and March, in contrast to 1% in BL. Abundance of quad-modal structures decreased to 3% in FT during December–February. Considering high abundance of the quad-modal structures in the air masses from FT in September–October (Fig. 5e), spring NPF might occur dominantly in FT. However, abundance of tri-modal and quad-modal structures with air masses of MBL and coastal BL increased (27–52%, mean 41%) during December–March. Abundance of quad-modal structures in BL increased to 7% in BL (3% in FT) during December–February. In addition to NPF in FT, the high abundance in BL during December–March implies that more NPF proceeded in the BL during summer, as reported from earlier works (Koponen et al., 2003; Virkkula et al., 2007; Asmi et al., 2010; Kyrö et al., 2013; Weller et al., 2015; Jokinen et al., 2018; Kim et al., 2019; Jang et al., 2019; Lachlan-Cope et al., 2020). It is noteworthy that the abundance of air masses from continental FT and coastal FT during summer decreased remarkably, even in general features (Fig. 5a). Furthermore, the quad-modal structure was observed in air masses from continental FT and coastal FT in December (summer). CN enhancement by NPF and growth was observed in the lower FT over Syowa Station during summer (Hara et al., 2011a). Therefore, the difference of contributable air mass origins in quad-modal structures between spring–autumn and summer might reflect not only the locations of NPF occurrence but also seasonal features of general air mass origins (Fig. 5a). Consequently, NPF might occur dominantly in FT and partly in BL during spring and autumn, and in BL and FT during summer.

### 3.3 Seasonal variations of aerosol physicochemical properties

Figure 7 depicts seasonal variations of all of the following: (a) concentrations of major aerosol constituents in $D_p$ <200 nm; (b) CN concentrations; (c) modal sizes and number concentrations of each mode; (d) aerosol number concentrations of fresh nucleation mode ($D_p$ = 5–10 nm); (e) coagulation sinks and condensation sinks; (f) nucleation rates of aerosol particles with $D_p$ = 5 nm ($J_5$); (g) extent of the Antarctic ozone hole; and (h) UV radiation near the surface at Syowa Station. Major water-soluble aerosol constituents of less than $D_p$ = 200 nm are $CH_3SO_3^-$ and non-sea-salt (nss) $SO_4^{2-}$ in summer and sea-salt (e.g., $Na^+$) in winter (Fig. 7a). The $CH_3SO_3^-$ concentrations, which start increasing at the end of August–September, show maximum concentrations in February–March at Syowa Station. Seasonal variation of $CH_3SO_3^-$ concentrations implies that oceanic bioactivity and atmospheric photochemistry contribute to the maintenance of aerosol systems in September – early April because precursors of $CH_3SO_3^-$ (e.g., DMS) derive from oceanic bioactivity (e.g., Minikin et al., 1998; Enami et al., 2016).

Variations of CN concentrations (Fig. 7b) exhibited clear seasonal features with a minimum in winter and a maximum in summer. During winter, CN concentrations increased occasionally under storm and strong wind conditions. Increase of CN concentration from winter minimum started in the end of August – early September at Syowa. CN concentrations and seasonal variations were similar to those measured at other coastal stations (e.g., Weller et al., 2011; Fiebig et al., 2014).

Our measurements show size distributions of ultrafine aerosol particles with mono-, bi-, tri-, and quad-modal structures (Fig. 2). The presence of tri-modal and quad-modal structures during spring–autumn suggests a frequent occurrence of NPF and growth in the atmosphere. Indeed, fresh-nucleation and aged-nucleation modes ($D_p$: <10 nm and 10–25 nm) appeared often in

spring–summer (Fig. 7c). Particularly, fresh nucleation mode appeared most frequently in the end-August – early October and February–March. Furthermore, the modal sizes increased gradually from fresh nucleation mode to the first Aitken mode ($D_p$ = 25–50 nm) during September–December. Such a gradual size shift implies that aerosol particles derived from NPF grew by the condensation of condensable vapors such as $H_2SO_4$.

Aerosol number concentrations of $D_p$ = 5–10 nm ($N_{5-10}$) show three maxima in September–October, December, and February–March (Fig. 7d) when tri-modal and quad-modal structures appeared frequently. These periods respectively correspond to the existence of the ozone hole (September–October marked by grey-shaded bands), the maximum of solar radiation (summer solstice marked by blue-shaded bands), and the minimum of the sea ice extent (February–March marked by green-shaded bands). In general, $N_{5-10}$ was controlled by (1) NPF and growth from smaller size (ca. 1 nm), (2) coagulation loss, and (3) particle growth to larger sizes by vapor condensation. Additionally, $J_5$ showed three maxima similar to $N_{5-10}$ variation (Fig. 7f). The estimated $J_5$ values from this study are comparable to the formation rates measured and estimated at Aboa (Kyrö et al., 2013), Neumayer (Weller et al., 2015), and Concordia (Järvinen et al., 2013), although a high formation rate was obtained in the Antarctic Peninsula (Kim et al., 2019). During this study, high values of $J_5$ were observed often during May – August. The high $J_5$ values corresponded to high CN concentrations and storm conditions during winter. Aerosol volatility measurements showed clearly that less volatile particles (i.e., sea-salt particles) were distributed in size ranges of $D_p$ >10 nm at Syowa during winter (Hara et al., 2011b). Indeed, sea-salt aerosols in ultrafine mode can be formed through sublimation of saline snow particles released from sea ice areas (Yang et al., 2019). High values of $J_5$ during winter might derive from dispersion of sea-salt aerosols from sea-ice areas by strong winds (Hara et al., 2011b). Therefore, high $J_5$ in the spring and summer are expected to be related to NPF.

The high values of $J_5$ and $N_{5-10}$ in December and February–March are explainable by photochemical reactions that form condensable vapors, and by the source strength of aerosol precursors released from oceanic bioactivity near the sea–ice margin. Indeed, high concentrations of DMS released from oceanic bioactivity were observed near sea-ice margins (Koga et al., 2014). By contrast, the sea–ice extent showed a maximum value in spring (September–November, Fig. 7g). Consequently, other factors affecting the likelihood of high $J_5$ and $N_{5-10}$ in spring should be considered. To elucidate the presence of fresh nucleation mode in spring, the location at which NPF occurred in the Antarctic region must be discussed. Air masses having quad-modal structures with fresh nucleation mode originated mostly from the upper FT over the Antarctic continent (Supplementary, Fig. S1). *Coag.S* of $D_p$ = 5 nm aerosol particles increased gradually from $2 \times 10^{-6}$ s$^{-1}$ to $10^{-5}$ s$^{-1}$ during September–December (Fig. 7e). From *Coag.S*, we estimated the e-folding time of the particles ($D_p$ = 5 nm) by coagulation loss as 3.9–5.8 days in early September and 1.2–1.4 days in December (Supplementary, Fig. S2). This finding implies that the particles in the fresh nucleation mode observed at Syowa had too short a lifetime to be supplied by long-range transport from the mid-latitudes via the FT. Therefore, NPF can be expected to start in the Antarctic FT already by the end of August.

Generally speaking, NPF is more likely to occur under conditions with (1) lower number concentrations of preexisting particles, (2) higher concentrations of condensable vapors, and (3) presence of sufficient photochemical oxidants such as OH. Because of low aerosol number concentrations in the Antarctic FT (Hara et al., 2011a), NPF can proceed preferentially in the FT if condensable vapor is present. Earlier studies presented that the following condensable vapors participate in tropospheric NPF: $H_2SO_4$, $CH_3SO_3H$, $NH_3$, iodine species such as $HIO_3$, and amines (Yu et al., 2012; Kulmala et al., 2013; Kyrö et al., 2013; Weller et al., 2015; Sipilä et al., 2016; Jen et al., 2016; Jokinen et al., 2018; Shen et al., 2019; Burrell et al., 2019). Actually, these vapors were identified along the Antarctic coasts and the Southern Ocean (Jefferson et al., 1998a, 1998b; Sipilä et al., 2016; Jokinen et al., 2018; Dall'Osto et al., 2019). High contributions of photochemical processes in the atmosphere were required for an important relation among $J_5$, aerosol size distributions, and CN concentrations. In addition, condensable vapors

can be removed by condensation onto pre-existing particles. Therefore, the condensable vapors must be formed in FT if NPF occurs in FT. Particularly, $H_2SO_4$, $CH_3SO_3H$, and iodine species such as $HIO_3$ are converted from precursors via photochemical reactions (e.g., Enami et al., 2016; Saiz-Lopez et al., 2008, 2011, 2014, 2016). Direct measurements of these vapors and their precursors in the Antarctic FT have never been reported.

Atmospheric iodine cycles are related closely to snowpack chemistry in the Antarctic (e.g., Atkinson et al., 2012; Roscoe et al., 2015; Saiz-Lopez et al., 2015; Hara et al., 2020). Because of fast reactions of reactive iodine species ($I_xO_y$; Saiz-Lopez et al., 2008; Atkinson et al., 2012; Roscoe et al., 2015), atmospheric chemical processes of $I_xO_y$ proceed dominantly near the surface (below 100 m). Hara et al. (2014) showed that significant injection of sea-salt aerosols originated from sea-ice areas into FT by cyclone activity. In addition, sea-salt aerosols originated from sea-ice area were distributed widely in ultrafine–
coarse particles at Syowa during winter–spring (Hara et al., 2011b, 2013, 2020). Considering that sea-salts in snow and frost flowers on seasonal sea-ice had iodine enrichment by sea-salt fractionation (Hara et al., 2017), the rapid vertical mixing of sea-salt particles as shown by Hara et al. (2014) plays an important role to supply iodine (mostly iodide) in aerosols into FT. Heterogeneous reactions on sea-salt particles containing iodine during transport can engender $I_xO_y$ release and formation of the iodine condensable vapors such as $HIO_3$. High concentrations of sea-salt particles, however, might prevent NPF by the
iodine condensable vapors in FT because of efficient condensation losses of vapors onto sea-salt aerosols.

DMS measurements taken at Concordia suggest that DMS was transported from coastal areas (or oceans) to the inland station via FT (Preunkert et al., 2008). Therefore, photochemical reactions with DMS, $H_2SO_4$, and $CH_3SO_3H$ might occur in the FT over the Antarctic continent. Sufficient photochemical oxidants such as OH and UV radiation are necessary for the conversion
of condensable vapors such as DMS and $SO_2$. Appearance of $O_3$ hole in the Antarctic stratosphere can engender UV enhancement with wavelengths shorter than 310 nm, even in the Antarctic troposphere during September–November. Therefore, atmospheric OH in the Antarctic troposphere is producible by UV radiation ($\lambda \leq 310$ nm) (Matsumi et al., 2002; Matsumi and Kawasaki, 2003).

Polar sunrise in the upper FT occurs earlier than it does near the surface. Additionally, the existence of the ozone hole enhances UV radiation, even in the troposphere, during September–November (Figs. 7g-h). More noteworthy is the higher UV of wavelength shorter than 305 nm in October–November than in December near the surface at Syowa. For example, the monthly mean amounts of UV radiation of 300–305 nm at Syowa were, respectively, 0.080, 0.098, and 0.068 kJ m$^{-2}$ in October, November, and December, respectively. In general, light intensity including UV radiation is greater at higher altitudes because
of atmospheric scattering and absorption. Considering the greater amount of UV radiation in the upper troposphere during September – November, the existence of the ozone hole might enhance the formation of photochemical oxidants (e.g., OH) and condensable vapors, which then engender NPF in the upper troposphere (Figs. 7c-d).

### 3.4 Vertical variations of aerosol e-folding time by coagulation loss

After NPF, aerosol particles in fresh nucleation mode are grown mainly by condensation of condensable vapors. They are coagulated efficiently onto pre-existing particles. For a better understanding of the aerosol life cycle including microphysical processes in the Antarctic troposphere, aerosol lifetime must be discussed before discussion of the relation between NPF and CCN ability. Williams et al. (2002) reported that aerosol particles of $D_p$ <60 nm were removed dominantly by coagulation in FT. Therefore, vertical variations of e-folding time were estimated as aerosol lifetime in the Antarctic FT from $Coag.S$ in each
size (Fig. 7e and Fig. S2). Aerosol size distributions in $D_p$ <100 nm remain unknown in the Antarctic FT through the year. Here, the e-folding time in FT was estimated, assuming that aerosol mixing ratio of aerosol number in respective size bins in

FT was equal to that measured at the surface. Aerosol size distributions in the air masses from coastal FT and continental FT and monthly mean vertical profiles of air temperature measured Japan Meteorological Agency (URL: https://www.data.jma.go.jp/antarctic/datareport/index-e.html) were used for this estimation.


Figure 8 depicts vertical variations of the e-folding time of aerosol particles with sizes of $D_p = 1$–50 nm in September–December. In general, smaller aerosol particles have a greater diffusion coefficient, so that the e-folding time was shorter, particularly in $D_p$ <10 nm. In other words, aerosol particles larger than $D_p = 20$ nm had a longer e-folding time. Because of lower air temperature at higher altitudes, the estimated e-folding time tended to increase at higher altitudes for all sizes and

months. The e-folding time at the upper troposphere (ca. 8.3 km, 300 hPa) was 3–4 times longer than that at the surface. When aerosol particles were grown to sizes larger than $D_p = 20$–30 nm, the e-folding time reached >30 days in the middle and upper FT. In estimation of aerosol lifetime by Williams et al. (2002), the aerosol lifetime was approximately 35 days in the upper FT and approximately 15 days in the middle FT in the INDONEX campaign. It must be noticed that these lifetimes were calculated by Williams et al. (2002) from aerosol concentrations higher than those at the surface of Syowa, and that the e-folding time in

Fig. 8 was estimated based on a constant mixing ratio and based on aerosol size distributions measured at the surface. Tethered balloon measurements at Syowa Station (Hara et al., 2011) presented a vertical gradient of CN concentrations that was lower in FT (up to ca. 2500 m) except for an aerosol-enhanced layer in lower FT. Additionally, aerosol number concentrations of particles with $D_p$ > 300 nm in upper FT over Syowa were 1–2 orders lower than those at the surface (Kizu et al., 2010). With a vertical gradient of the number concentrations of pre-existing particles, the e-folding time of 1–2 months might be available

and realistic for aerosol particles with size of $D_p$ > 20–30 nm in the Antarctic FT.

In contrast to the longer e-folding time in FT, the e-folding time was shorter in BL (below 850 hPa) because of efficient coagulation loss. Particularly, the e-folding time at the surface was shorter in air masses originated from MBL and coastal BL than that in FT (not shown) because of high concentrations by primary emission of sea-salt aerosols in strong winds and NPF

in MBL and coastal BL. In other words, a high growth rate (i.e., high concentrations of condensable vapors) was necessary for particle growth for shorter periods (within the e-folding time) in BL. For example, under conditions with growth rates of 0.05, 0.1, 0.2, and 0.5 nm h$^{-1}$, it takes approximately 3.3 days, 1.7 days, 0.8 days, and 0.33 days, respectively, for initial growth from $D_p = 1$ nm to 5 nm. Under the slow growth conditions, the time for initial growth exceeded the e-folding time of $D_p = 1$ and 5 nm in BL. In FT, however, aerosol particles can continue to exist for a longer time under conditions with slower particle growth

(lower concentrations of condensable vapors) than those in BL. Consequently, aerosol particles can be grown gradually through condensation of condensable vapors in the Antarctic troposphere once new particles grow to the aged-nucleated particles ($D_p$ >10 nm). Finally, UV enhancement in the upper troposphere by the ozone hole during the end of August through November might modify the aerosol population and size distributions in the Antarctic troposphere during spring–summer. Therefore, aerosol properties in the Antarctic troposphere during spring–summer are not "pristine" but under Anthropocene conditions,

although the Antarctic troposphere remains the cleanest on the earth.

### 3.5 CCN potential and cloud amounts during summer

Gradual change of modal sizes in fresh nucleation mode and first Aitken mode were observed during spring–summer (Fig. 7c). When aerosol particles derived from NPF during periods with existence of $O_3$ hole grow to a critical diameter, the aerosol particles can act as CCN. Here, aerosol particles grown after NPF during periods with $O_3$ holes were specifically examined to

elucidate their relation to CCN ability and CCN properties during summer. As shown in Figure 7c, modal sizes increased gradually in aged-nucleation mode and first Aitken mode from the end of August through December. This gradual feature of the modal sizes was likely to be associated with particle growth after NPF. Measurements of aerosol hygroscopicity at Aboa showed the critical diameter as approximately ca. 50 nm during summer (Asmi et al., 2010). Considering gradual particle

growth in the aged-nucleation and first Aitken modes (Fig. 7c), NPF during the periods with $O_3$ hole and growth are expected to be linked to cloud properties during summer. Therefore, we attempt to estimate the contributions of aerosols derived from NPF during periods with $O_3$ hole to act as CCN in November–January based on the following assumptions: aerosol particles supplied from spring NPF were grown to aged-nucleation and first Aitken modes. For this estimation, the contribution to CCN ($R_{CCN}$) was defined as the ratio of the aerosol number concentrations of $D_p > 50$ nm in aged-nucleation and first Aitken modes ($N_{D>50}$) to the total aerosol number concentrations in $D_p > 50$ nm (total-$N_{D>50}$, Fig. 9). The value of $R_{CCN}$ was estimated using the following equation.

$$R_{CCN}(\%) = \frac{N_{D>50}}{total-N_{D>50}} \times 100 \quad (7)$$

Figure 10 depicts $R_{CCN}$ in the continental FT and coastal BL in November–January. Results obtained using ANOVA test indicated a significant difference ($p = 0.0125$) between the values of $R_{CCN}$ obtained in the continental FT and coastal BL. Differences of aerosol size distributions are regarded as indicating $R_{CCN}$ in the continental FT and coastal BL. As shown in Figure S3, the modal size in the smallest mode was smaller in BL than those in FT in December–January, although the modal size was not significantly different in November. Aerosol particles of $D_p < 30$ nm were supplied by NPF and growth, as described above. Therefore, NPF in the BL in December–January might engender a difference of modal sizes in the smallest mode. The modal size in the smallest mode in BL was smaller than the critical diameter. Therefore, aerosol particles supplied from summer NPF in the BL were not grown sufficiently to the critical diameter by condensation. We infer that aerosol enhancement by NPF during periods with the existence of $O_3$ hole might modify not only aerosol size distributions but also CCN ability and cloud properties in summer.

Trends of cloud amounts at Syowa Station were examined to assess the effects of spring NPF enhanced by the existence of $O_3$ hole on CCN ability and cloud properties during summer. Figure 11 depicts variations of cloud amounts in December and January that have been recorded since 1969. No increasing trend was found for air temperature or water vapor concentrations related to the cloud amount, but a marked increasing trend with $p$ values lower than 0.01 was found. Nevertheless, it was found only for December and January during 1969–2012. Particularly, cloud amounts at Syowa Station were significantly greater during the existence of $O_3$ hole (Fig. 11b, and Fig. S4). Consequently, the cloud amount trend during summer might result from spring NPF enhanced by $O_3$ hole existence. Because of the conformable relation between the cloud amount and radiative fluxes (Yamanouchi et al., 2007), UV enhancement by the $O_3$ hole can affect atmospheric radiation budgets by aerosol and cloud properties in the Antarctic during summer.

Our hypothesis can be summarized as shown in the schematic figure (Fig. 12). In polar sunrise, solar radiation recovers earlier in the stratosphere and the upper troposphere. Then, ozone depletion by catalytic reactions of chlorine cycle starts in the Antarctic stratosphere. The ozone hole appears in the Antarctic stratosphere from the end of August until the end of November. Appearance of the ozone hole engenders UV enhancement and then production of atmospheric oxidants such as OH in the troposphere. Atmospheric oxidants such as OH are likely to be formed in the upper troposphere by UV enhancement. Condensable vapors (i.e., aerosol precursors with lower vapor pressure) are producible by photochemical oxidation. $H_2SO_4$ and $CH_3SO_3H$ are important condensable vapors for NPF and growth in the Antarctic FT. In addition, NPF occurs already by the end of August in the upper troposphere. As a consequence, NPF is enhanced in FT under UV enhancement by $O_3$ hole during spring (end of August – November). Aerosol particles in fresh nucleation mode grow gradually by the condensation of condensable vapors in the FT. Some aerosol particles in fresh and aged nucleation modes are transported to the lower troposphere in the Antarctic coasts. With condensation growth, some aerosol particles originated from spring NPF in the FT can be grown to sizes greater than the critical diameter (ca. 50 nm) in the summer. Results show that NPF proceeds even in

coastal BL and MBL from the end of spring – summer. In contrast to FT, $H_2SO_4$, $CH_3SO_3H$, and other condensable vapors such as $HIO_3$, $NH_3$, and amines can engender NPF and growth in BL. Aerosol particles originating from NPF in BL are removed efficiently by coagulation under conditions with higher aerosol concentrations of pre-existing particles than those of FT, except under high growth rate conditions. Consequently, lesser aerosol particles derived from summer NPF in the boundary layer grow up to the critical diameter than from spring NPF in the FT. Our results demonstrate that spring NPF plays an important role in aerosol populations and cloud properties in the summer under conditions that include the existence of an Antarctic ozone hole. Knowledge of the following issues will be necessary for future investigations to test our hypothesis: (1) aerosol size distributions of particles smaller than 100 nm in the FT, (2) chemical forms of condensable vapors and their concentrations in the FT and BL, and (3) their vertical and seasonal variations in the Antarctic.

**4. Conclusion**

Aerosol measurements were taken using SMPS and OPC at Syowa Station, Antarctica in 2004–2006. Aerosol size distributions were found to have mono-, bi-, tri-, and quad-modal distributions during our measurement period. The mono-modal distribution was dominant under strong wind conditions during May–August. The bi-modal distribution was identified through the year. Tri-modal and quad-modal distributions were observed mostly during September–April. Seasonal features of $N_{5-10}$ and $J_5$ imply that NPF identified at Syowa Station was associated with UV enhancement by ozone hole existence in spring, maximum of solar radiation in summer, and minimum of sea-ice extent in February–March. In addition, NPF occurs in FT during spring and autumn and in the FT and boundary layer during summer. Additionally, spring NPF and particle growth are linked to cloud properties during summer.

We obtained direct evidence indicating that spring UV enhancement by the ozone hole engendered spring NPF and growth in the Antarctic FT. With ozone hole recovery (Solomon et al., 2016; Kuttippurath et al., 2017), aerosol properties and populations might be modified for the next several decades. Consequently, indirect effects on atmospheric radiation budgets and climate change in the Antarctic regions during the summer can revert to levels that prevailed before the ozone hole existence. More aerosol measurements must be taken in Antarctic regions to monitor these trends and future effects.

**Data availability**

Data are available by contacting the corresponding author (KH: harakei@fukuoka-u.ac.jp).

**Author contributions**

KH, KO, and TY designed aerosol measurements at Syowa Station. KH, KO, and MY conducted wintering aerosol measurements at Syowa Station in 2004–2006. KH and CNH contributed to data analysis including SMPS/OPC data and meteorological data. KH prepared the manuscript and led data interpretation. All co-authors contributed to discussions about data interpretation and the manuscript.

**Competing interests**

The authors declare that they have no conflict of interest.

**Acknowledgements**

We thank the members of the 45th–47th Japanese Antarctic Research Expedition for assistance with aerosol measurements taken at Syowa Station. This study was supported financially by the "Observation project of global atmospheric change in the Antarctic" for JARE 43–47. This work was also supported by Grants-in Aid (No. 16253001, PI to T. Yamanouchi; No. 15310012, PI to K. Osada; and No. 22310013, PI to K. Hara) from the Ministry of Education, Culture, Sports, Science and Technology of Japan. This study was supported by National Institute of Polar Research (NIPR) through Project Research no. KP-302. The authors gratefully acknowledge the NOAA Air Resources Laboratory (ARL) for providing the HYSPLIT Transport and Dispersion Model and READY website used for this research (http://www.arl.noaa.gov/ready.html).

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

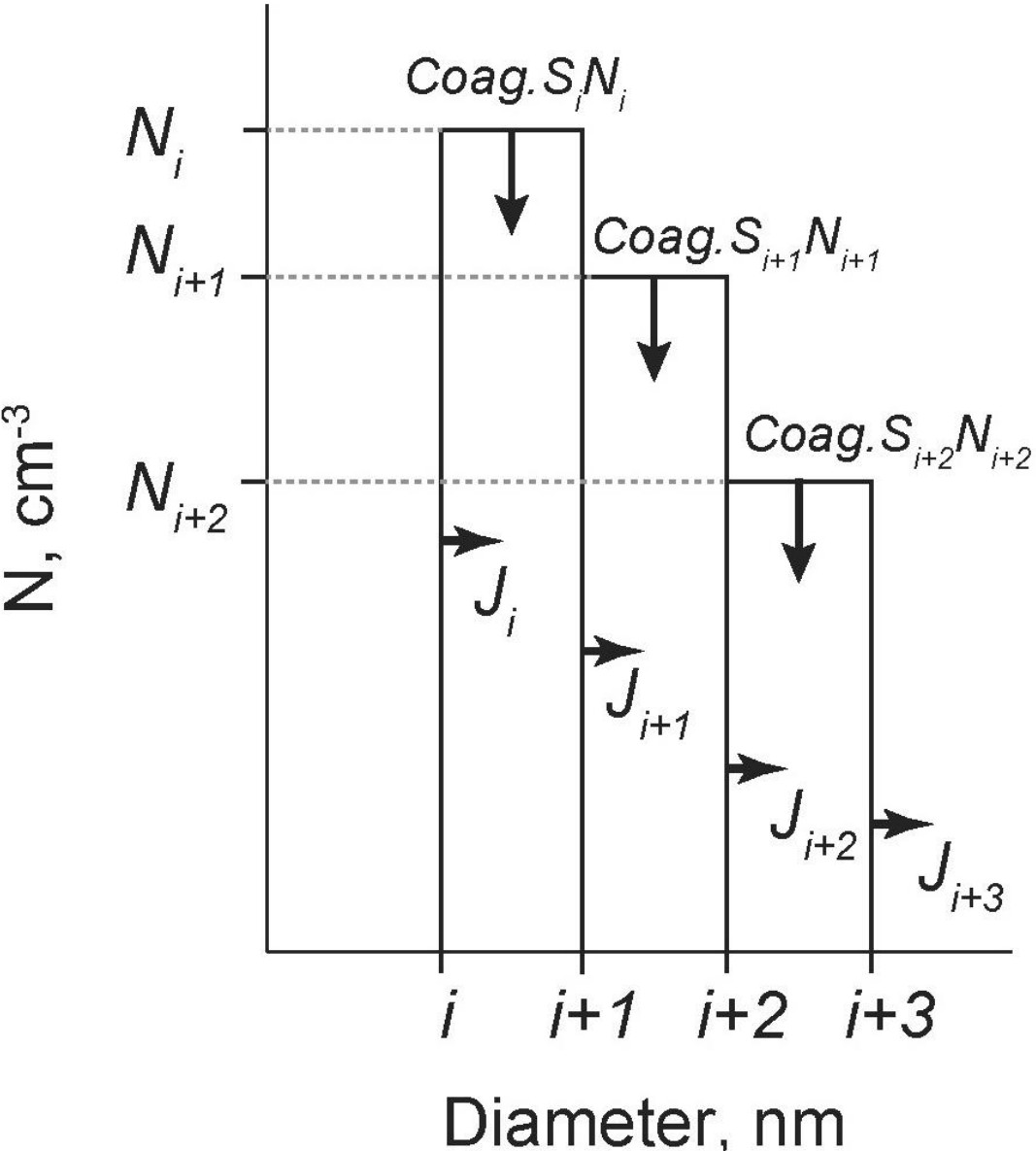

**Figure 1: Schematic figure showing procedures used for the nucleation rate of aerosol particles with size of $D_p$ = 5 nm (J$_5$). $N_i$, $Coag.S_i$, and $J_i$ respectively indicate the number concentrations, coagulation sink, and growth (or formation) rate for each size bin.**


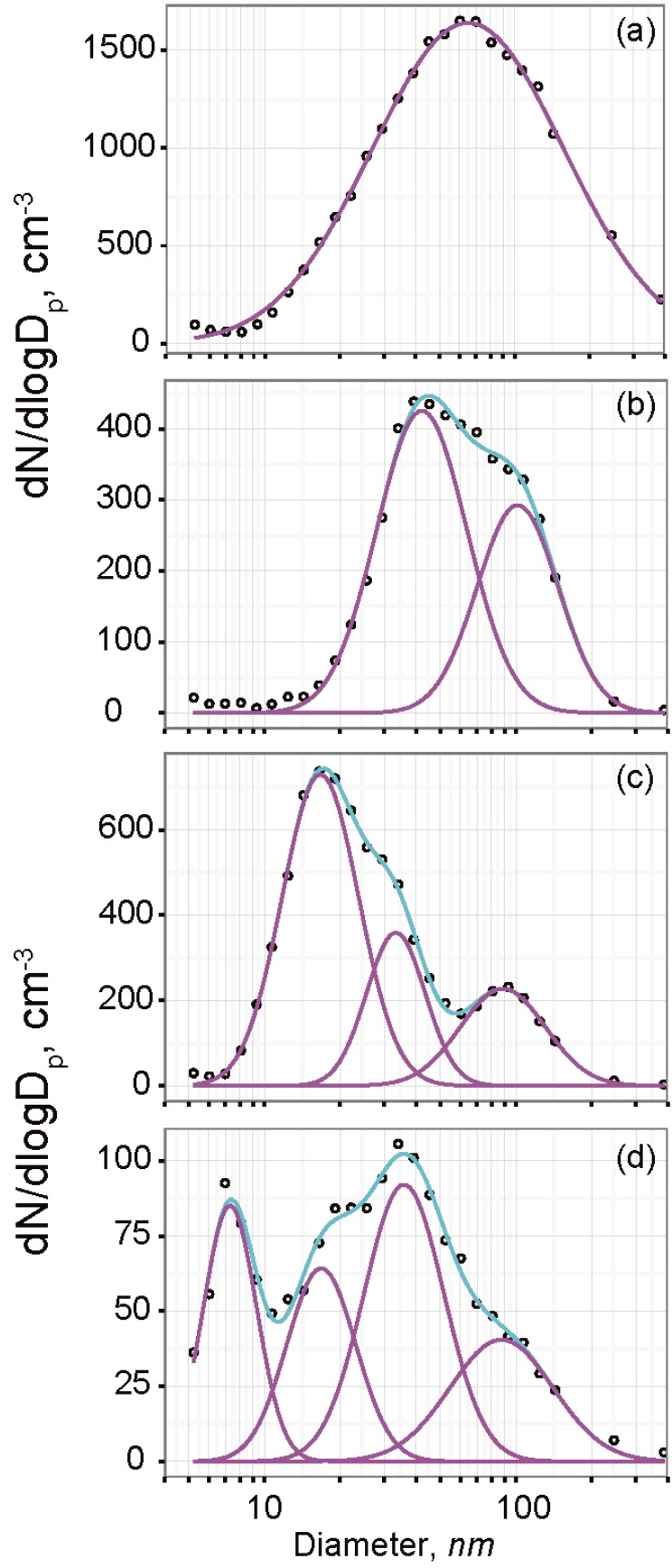

**Figure 2: Examples of aerosol size distributions with (a) mono-modal (3 May 2005), (b) bi-modal (4 March 2005), (c) tri-modal (14 February 2005), and (d) quad-modal (8 September 2006) structures. Circles, pink lines, and cyan lines in (a–d), respectively show the observed data by SMPS, the number concentrations in each mode by approximated by lognormal fitting, and total concentrations of the respective modes.**

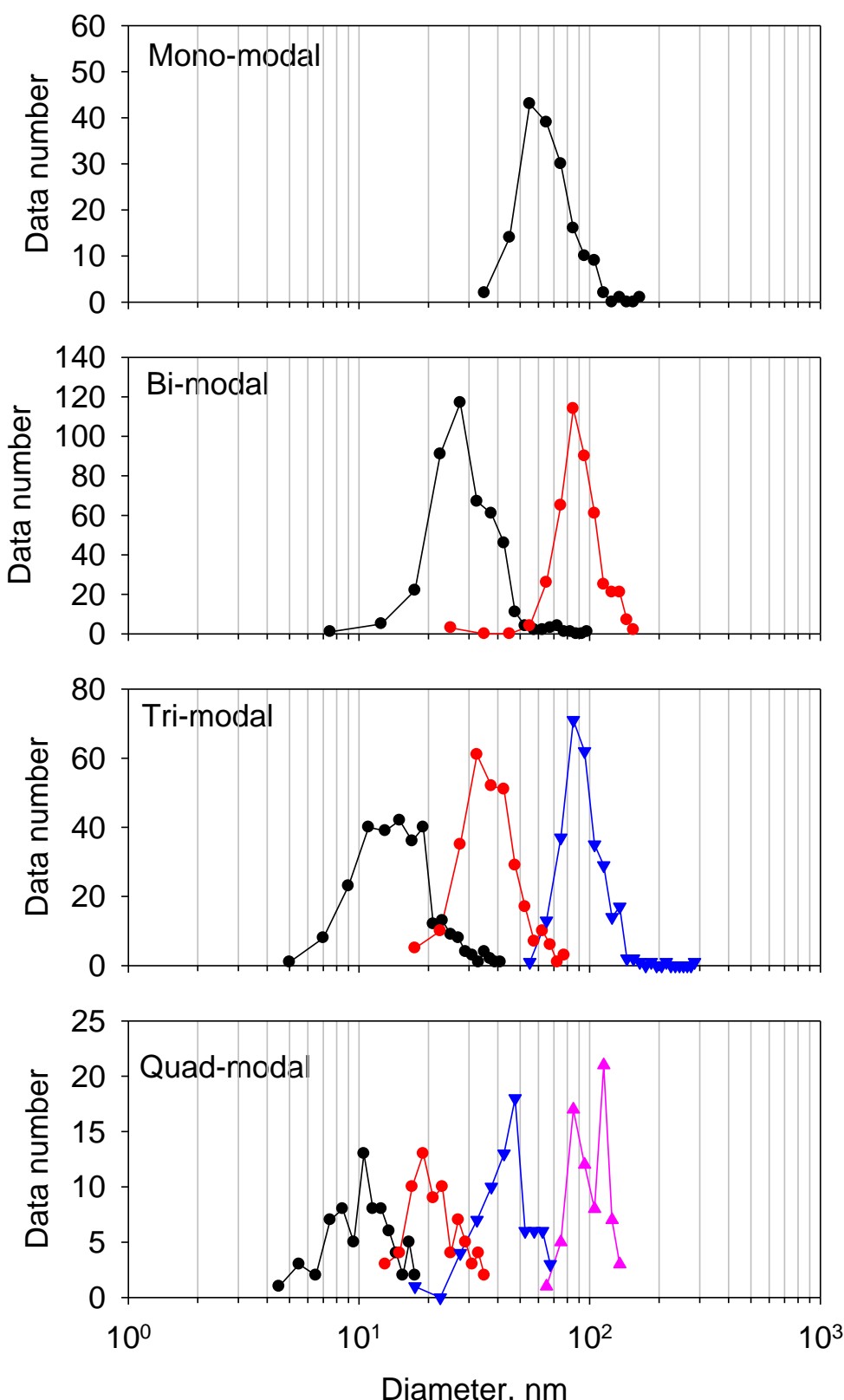

**Figure 3: Histogram of modal sizes in each modal structure. Symbols and lines of black, red, blue, and magenta show histograms of first, second, third, and fourth modes in each modal structure.**

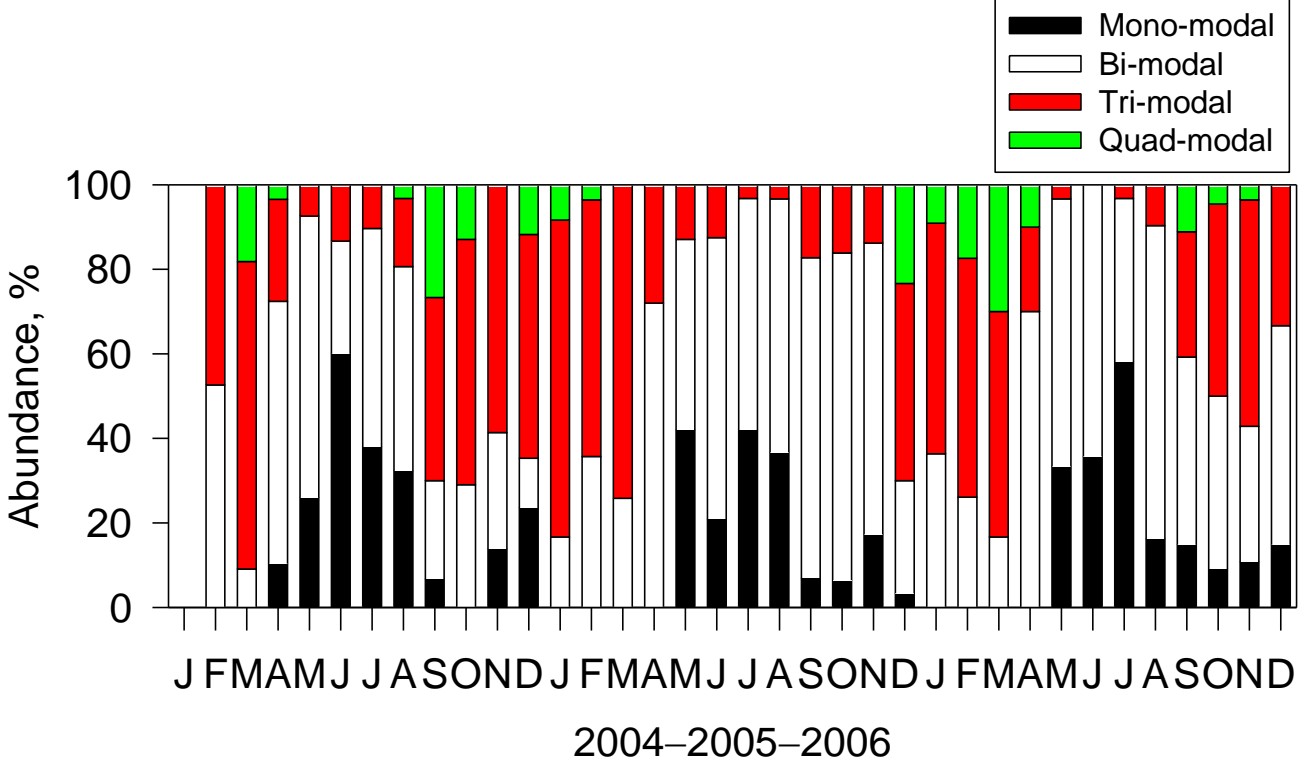

**Figure 4: Seasonal features of abundance of modal structures observed at Syowa Station, Antarctica.**

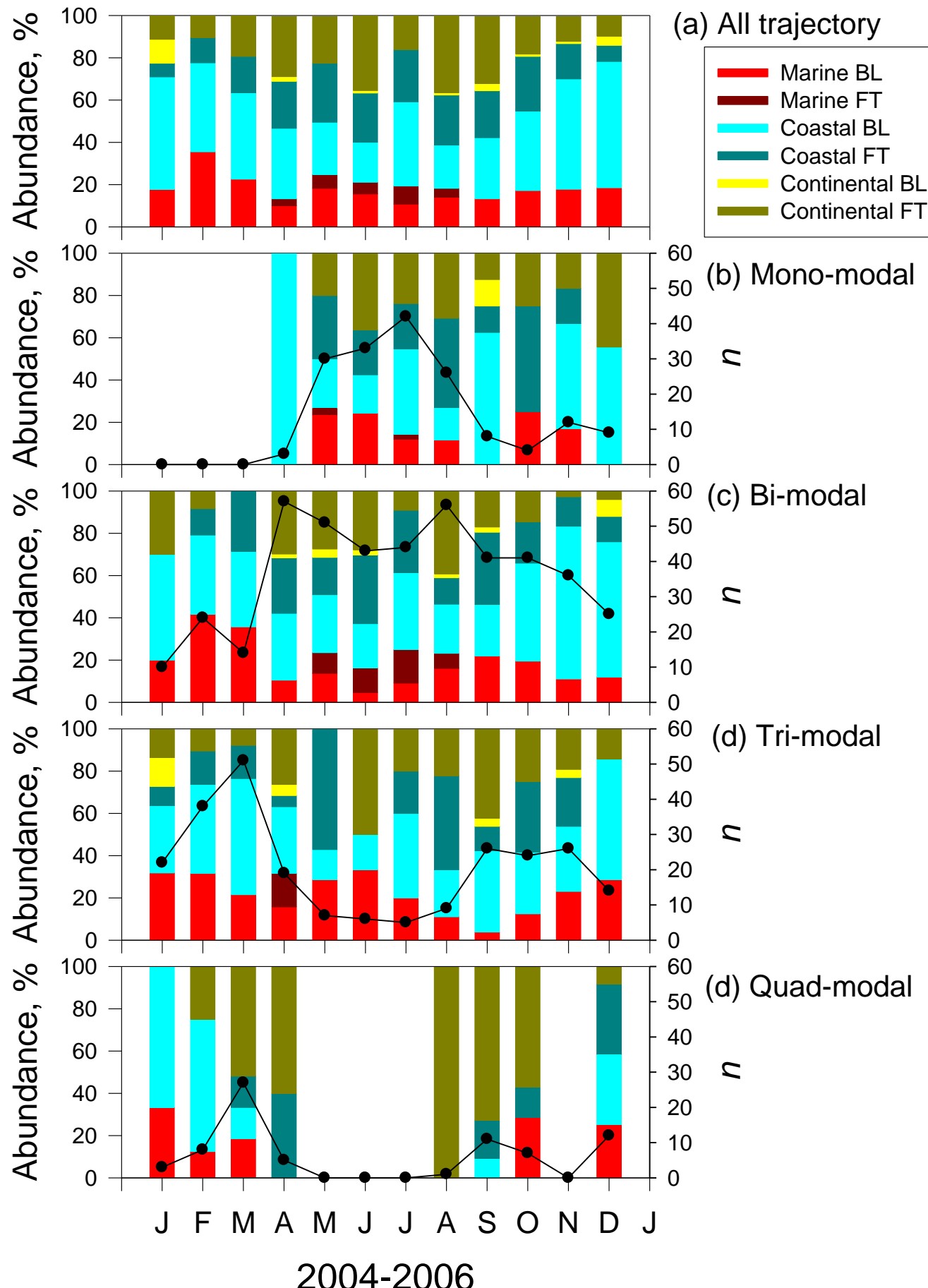

Figure 5: Seasonal feature of air mass origins of (a) all trajectory, (b) mono-modal structure, (c) bi-modal structure, and (d) tri-modal structure, and (e) quad-modal structure at Syowa Station, Antarctica. Black line, black circles, and "n" represent the number of the appearance of each modal structure.


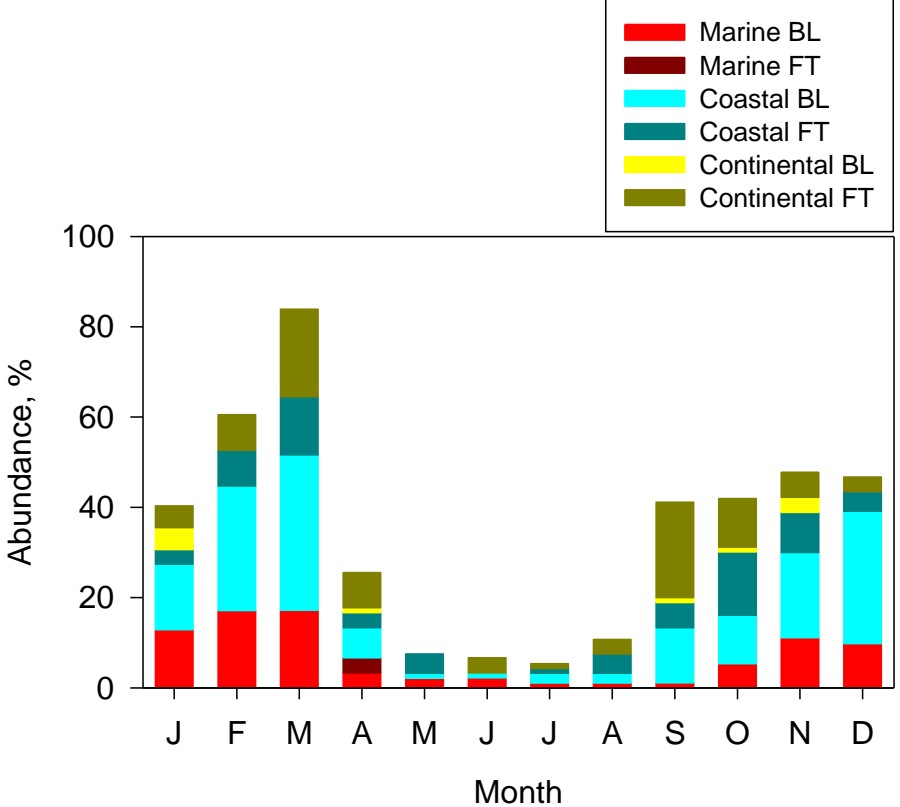

**Figure 6: Seasonal features of abundance of tri-modal and quad modal structures and air mass origins at Syowa Station, Antarctica in 2004-2006.**


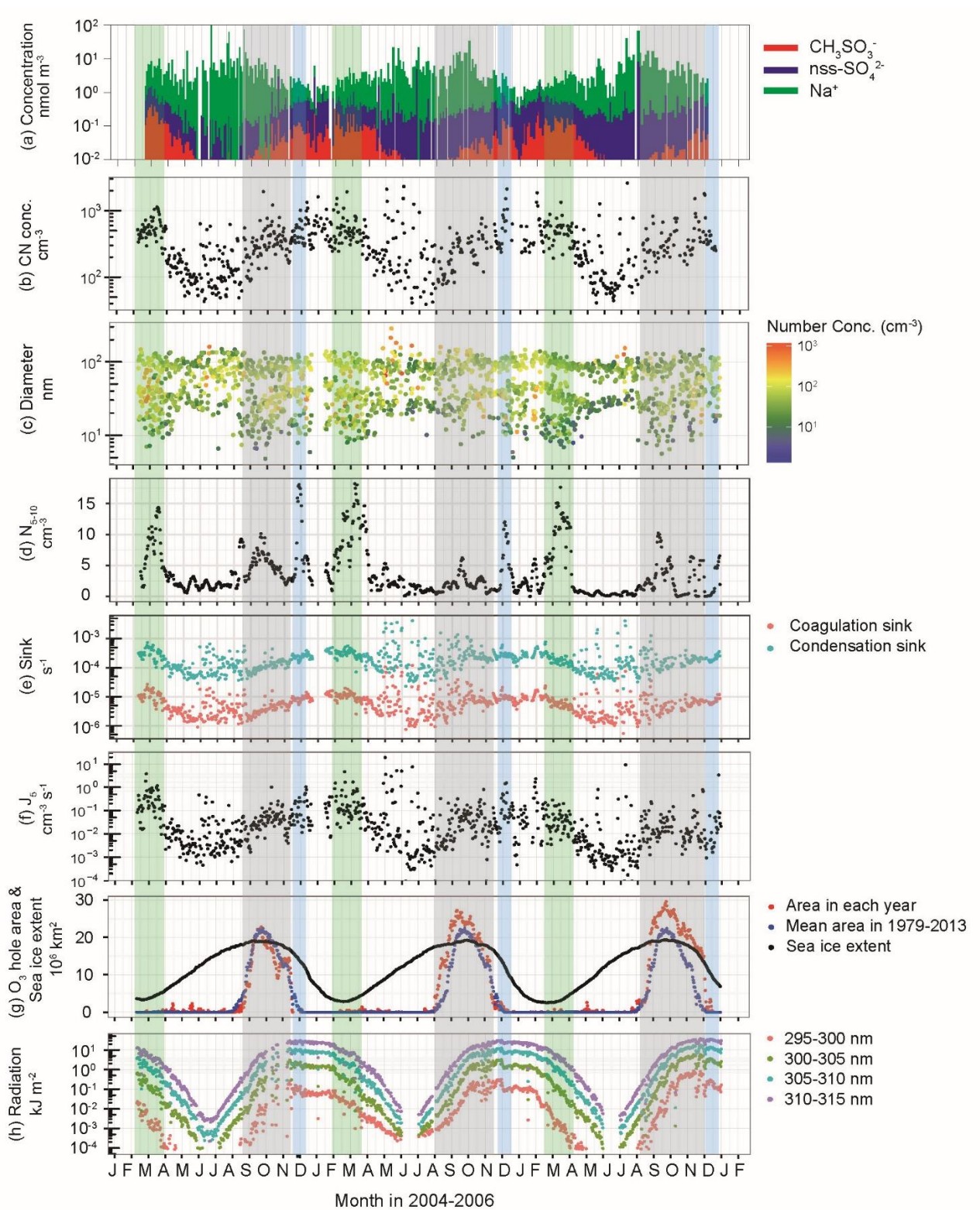

**Figure 7: Seasonal variations of (a) the concentrations of CH₃SO₃⁻, non-sea-salt (nss-) SO₄²⁻ and Na⁺ in $D_p$ <200 nm, (b) CN concentrations ($D_p$ >10 nm), (c) modal sizes and number concentrations in each mode, (d) 30-day running mean aerosol number concentrations of fresh nucleation mode ($D_p$ = 5–10 nm), (e) coagulation sink of aerosol particle ($D_p$ = 5 nm) and condensation sink, (f) nucleation rate of aerosol particles with $D_p$ = 5 nm ($J_5$), (g) extent of the Antarctic ozone hole, and (h) UV radiation near surface at Syowa Station, Antarctica during January 2004 – December 2006. Green-shaded, grey-shaded and blue-shaded bands respectively represent periods of minimum of sea-ice extent and high CH₃SO₃⁻ concentrations, ozone hole appearance and summer solstice. Concentrations of nss-SO₄²⁻ were calculated using Na⁺ concentrations and molar ratios in seawater (SO₄²⁻/ Na⁺ = 0.0602; Millero et al., 2008) during November–March and ratios in sea-salts (SO₄²⁻/Na⁺ = 0.01; Hara et al., 2012, 2018) in April–October because the SO₄²⁻/Na⁺ ratio is changed by sea-salt fractionation on sea-ice during April–October (Hara et al., 2012, 2018). The ozone hole extent and sea ice extent data were provided, respectively, by NASA (https://ozonewatch.gsfc.nasa.gov/) and the National Snow & Ice Data Center (https://nsidc.org/data/seaice_index). Daily mean UV data at Syowa Station were monitored by the Japanese Meteorological Agency (http://www.jma.go.jp/jma/index.html) using a Brewer spectrophotometer.**

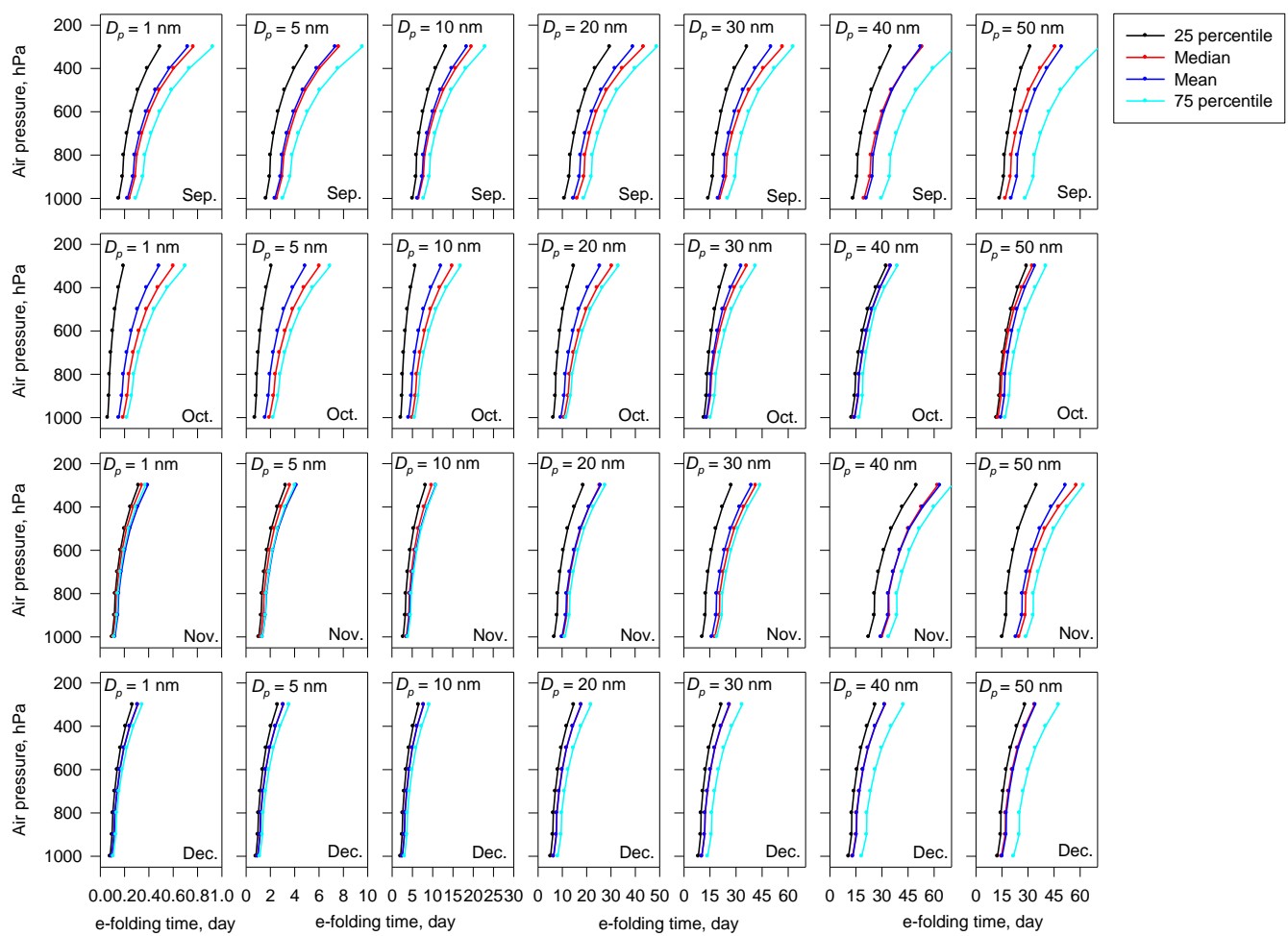

**Figure 8: Vertical variations of aerosol e-folding time of coagulation loss in respective sizes in the troposphere over Syowa Station, Antarctica during September–December.**

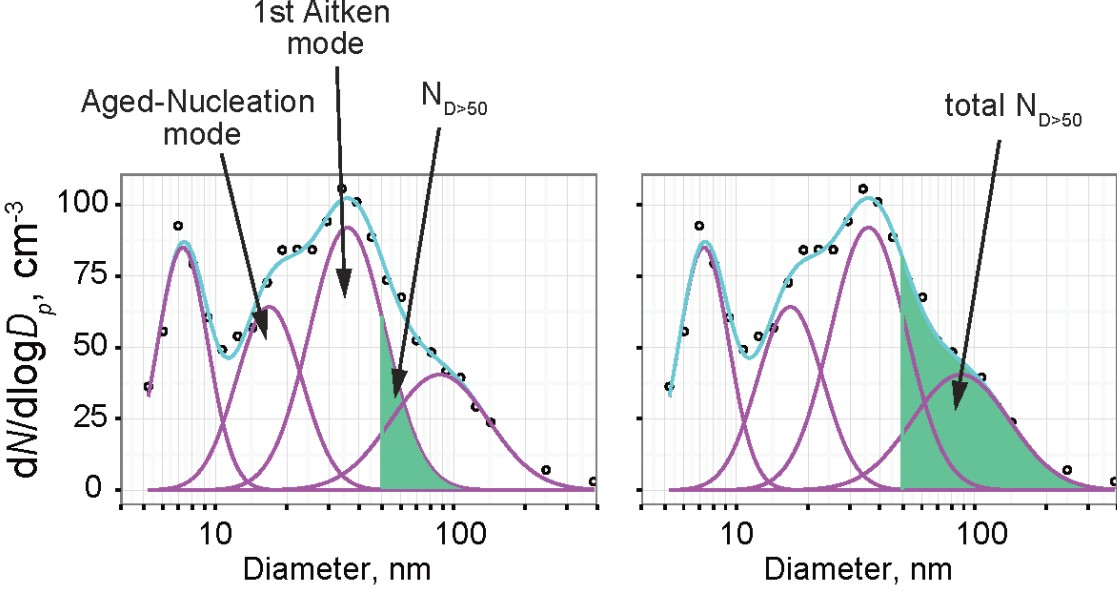

**Figure 9: Schematic figure showing procedures used for $R_{CCN}$ estimation. Circles, pink lines, and cyan lines respectively show the data observed using SMPS, the number concentrations in each mode by approximated by lognormal fitting, and total concentrations of each mode. Number concentrations of aerosols with size of $D_p$>50 nm in aged-nucleation mode and first Aitken mode ($N_{D>50}$) are given as shown for (a). Similarly, the total number concentrations of aerosols with size of $D_p$>50 nm (total-$N_{D>50}$) are given as shown for (b).**


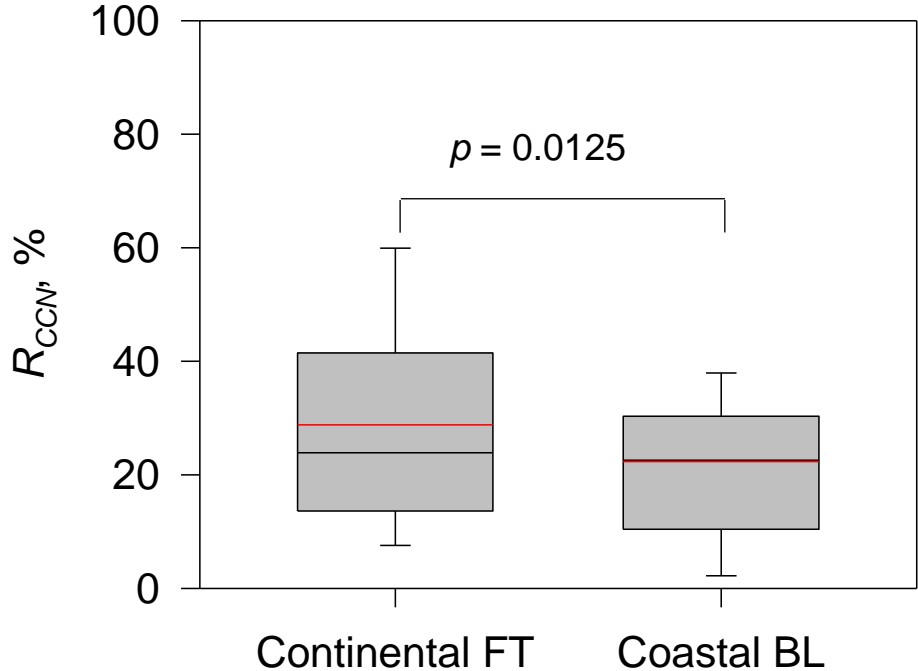

**Figure10: Comparison of R$_{CCN}$ in the continental free troposphere (FT) and those in the coastal boundary layer (BL) during November – January in 2004–2006. *p* denotes *p*-value of ANOVA variance test. Degrees of freedom for the ANOVA variance test were 155. Box plots show values of 90, 75, 50 (median), 25, and 10%, denoted respectively by the top bar, top box line, black middle box line, bottom box line, and bottom bar. Red lines show mean values.**


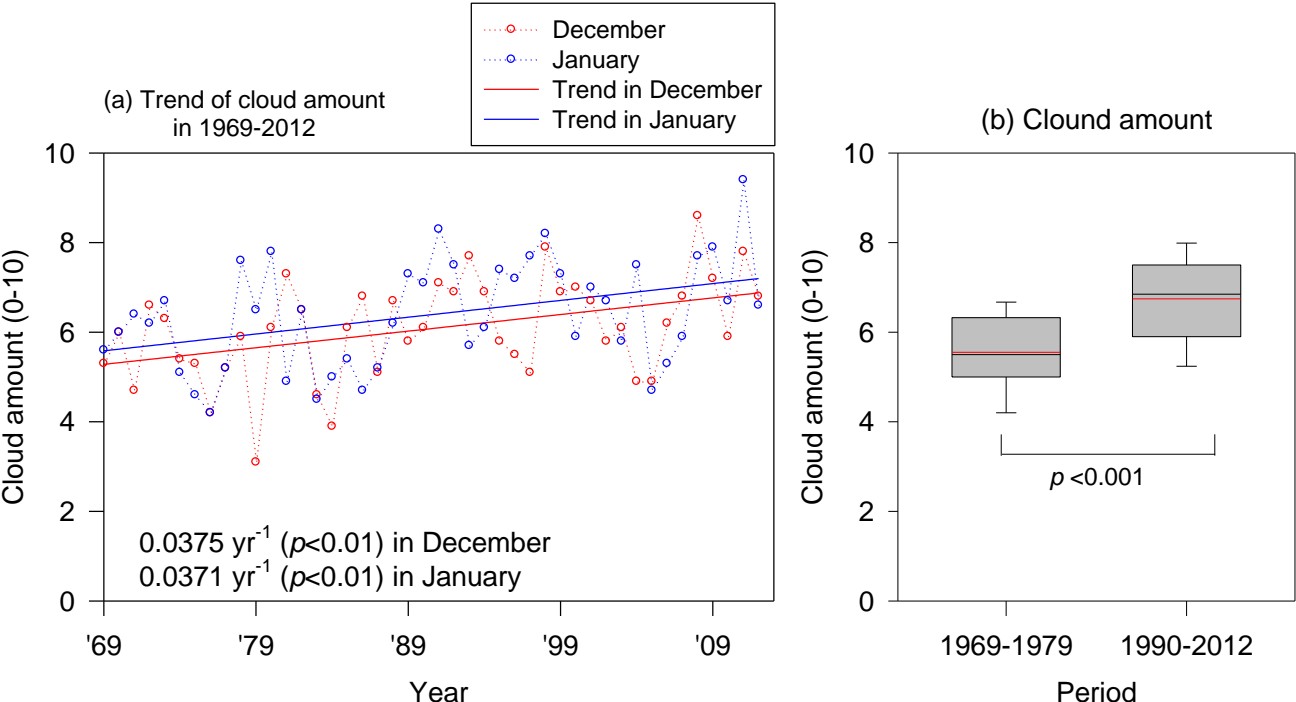

**Figure 11(a) Trend of monthly mean cloud amount in 1969–2012 and (b) comparison of cloud amounts in December–January before and after appearance of the Antarctic ozone hole at Syowa Station, Antarctica. *p* denote *p*-value of ANOVA variance test. Degrees of freedom for the ANOVA variance test were 66. Because of the extended period of the ozone hole in 1980–1989, cloud amount data in the period were excluded from Fig. 9b. Box plots show values of 90, 75, 50 (median), 25, and 10% denoted respectively by the top bar, top box line, black middle box line, bottom box line, and bottom bar. Red lines show mean values. Cloud amounts were observed**
**based on visual observations by the Japan Meteorological Agency (http://www.jma.go.jp/jma/index.html).**

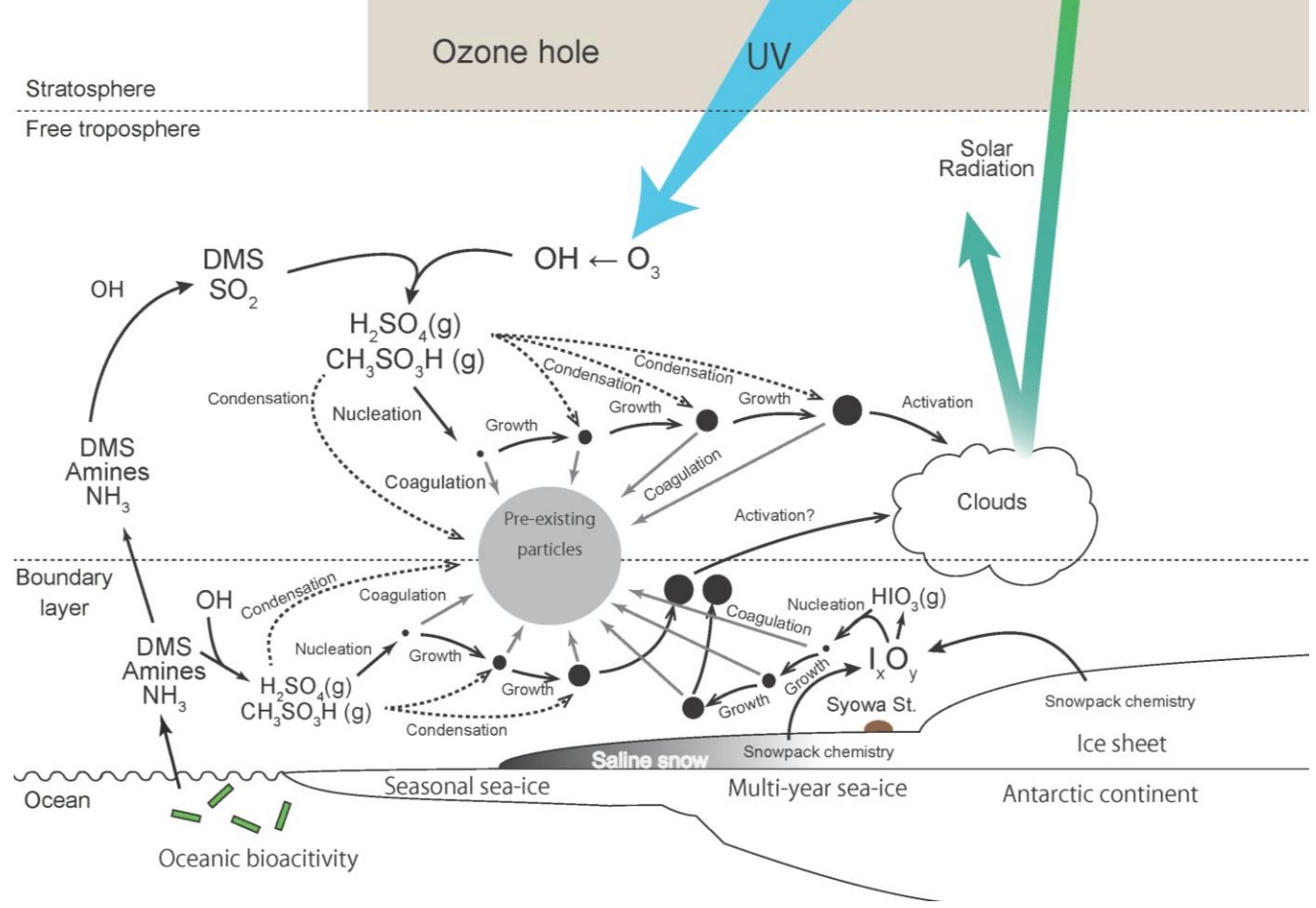

Figure 12: Schematic figure presenting our hypothesis.