# Peer review of "Characterization of aerosol number size distributions and their effect on cloud properties at Syowa Station, Antarctica"

_Atmospheric Chemistry and Physics, 2021_

## Author Comment (AC1)

Reply to comments from Reviewer#1:

We would like to thank your helpful comments to improve our manuscript. All comments are responded and addressed in the current revise. Details are listed as follows.

*Comment from Referee:* Unfortunately, the article in its present appearance has some weak points. It is regrettable that in several issues the reasoning is barely traceable and inadequately supported by the presented data evaluation. This is especially obvious concerning the claimed link between ozone hole -> NPF –> CCN –> cloud properties –> cloud amount. I think it is worth the effort addressing this weakness and considering a more in-depth analysis.

**Reply from Author:** When aerosol particles derived from NPF enhanced by ozone hole are grown to critical size for CCN activation, these particles can play important roles in the link. It is certain that model simulation is one of useful method to understand atmospheric processes as pointed by you, but we still have many uncertain parameters and knowledge such as aerosol size distributions, specific chemical forms of the condensable vapors, and the concentrations of the vapors in the Antarctic free troposphere. Because we must put many assumptions with high uncertainty into the model calculation, we do not use model simulation in the revise manuscript. Instead, we attempted to estimate aerosol lifetime from coagulation sink in the Antarctic free troposphere to ascertain this link among ozone hole, NPF, CCN ability, and cloud amount. Details are explained in each comment as follows.

*Comment from Referee:* Chapters 2.2.2, lines 112-119: To be honest, I do not understand the mathematics behind your estimate regarding J5. Why do you need the term Coag.S10-20N10-20, but then neglecting the condensation sink Cond5-10N5-10? Moreover, it is not even clear in which way you calculated the condensation sink shown in Figure 6d. Please specify input values for eq. (2) and in addition show the measured total particle concentrations in Figure 6.

**Reply from Author:** More explanation to estimate the formation rate of $J_5$ was added into the revised manuscript. In the revised manuscript, calculation procedures were in accordance with Dal Maso et al. (2002). Also, we added CN concentrations in the revised figure, which were measured simultaneously with SMPS at Syowa Station. Also, measurement procedure was added into the section of 2.1.

*Comment from Referee:* Chapters 2.3: Back trajectory analyses are a crucial tool in this study, so the authors should provide more details: Why did you rely on the NCEP meteorology

data set? The GDAS dataset has a higher resolution and is more accurate in general! Why an initial starting point of 500 m above ground has been chosen, well above the aerosol measuring point? With hysplit (using GDAS input), it is possible to start trajectory ensembles from different high levels. This option could be useful to assess the reliability of the back trajectory analysis. Another point: 5-days back trajectory may not be sufficient to address the origin of particles in the accumulation mode. Finally, does the chosen 1500 m boundary level between FT and BL refer to height above ground?

**Reply from Author:** It is true that GDAS dataset has a higher resolution than NCEP-reanalysis, but GDAS dataset is available since January 2005. In this study, the measurement period was February 2004 – December 2006, so that we used NCEP-reanalysis dataset to keep same quality during the measurement periods. As pointed by you, 5-day backward trajectory is slightly short (not sufficient) to know the origins of aerosol particles in the accumulation mode. Considering uncertainty of the trajectory analysis in troposphere and high uncertainty at lower altitudes, we set initial height of 500m corresponding to upper boundary layer over Syowa and calculation period of 5 days. In this study, we used results of the backward trajectory mostly to identify air mass origins and pathway for nucleation modes and modal structures. Therefore, we can compare the relation and identification by 5-day backward trajectory. To divide air mass origins, we chose altitude of 1500 m, because height of top of boundary layer was distributed below 1400 m through the year, which were observed by the tethered balloon measurements in 2005 at Syowa Station (Hara et al., 2011a, 2013).

*Comment from Referee:* Chapters 3.2, lines 180-181: Is there any evidence from your data that in this case sea-salt originated from the snow surface? This should be specified, otherwise a reference is needed.

**Reply from Author:** Direct evidence was already published in previous works (other papers, e.g., Hara et al., 2012, 2020). We added more explanation and the specific references about sea-salt aerosols originated from the snow surface in the revised manuscript. Simultaneous measurements of aerosol constituents showed that sea-salt aerosols in ultrafine – coarse modes were released from sea-ice area during winter – spring (e.g., Hara et al., 2012, 2020).

*Comment from Referee:* Chapters 3.2, lines 185-186: In this case, 10-days back trajectories could be beneficial!

**Reply from Author:** As mentioned above, we did not calculate 10-day backward trajectory.

***Comment from Referee:*** Chapters 3.3, line 223: "Particularly, fresh nucleation mode appeared only in end-August…" you mean fresh nucleation mode without aged nucleation mode?

**Reply from Author:** In the sentence of "Particularly, fresh nucleation mode appeared only in end-August…", we focused on fresh nucleation mode (D<10 nm), because fresh nucleation mode was direct evidence of recent NPF.

***Comment from Referee:*** Chapters 3.3, lines 259-264: Please delete this sentence as well as the (R1) and (R2), because it is (very) basic textbook knowledge.

**Reply from Author:** Reactions (R1 and R2) were removed in the revised manuscript. The sentences immediately before R1 and R2 were modified in the revised manuscript.

***Comment from Referee:*** Chapters 3.3, lines 266-267: This is not visible in Fig. 6g! I daresay that UV radiation is roughly comparable in Oct/Nov and December. Please provide numbers of the measured difference.

**Reply from Author:** Seasonal variation of solar radiation at Syowa (not shown in Figure) was clearly lower in October – November than that in December. Nevertheless, UV amount in October – November was higher or comparable than that in December. Especially, UV amount in shorter wavelength in October – November was obviously higher than that in December. We added short explanation and comparison of values of UV amounts in this period into the revised manuscript.

***Comment from Referee:*** Chapters 3.3, lines 270-279 and Chapter 3.4: This part is rather speculative and barely convincing! I agree that enhanced UV-radiation under ozone depletion conditions may potentially have an impact on NPF and CCN concentration during this period. However, in my view it is hard to believe that those NPF events and their subsequently grow to CCN relevant diameters will last more than around 2 months in the FT and will then have any significant influence on cloud properties in Dec/Jan! First, you estimated particle lifetime solely based on coagulation sink. Is there some evidence, that such a simplification is adequate? Moreover, consider that comparable photochemical processes provoking NPF in BL and FT surely also proceed during Dec/Jan (note comparable UV radiation and even more prominent DMS emissions). Thus, I cannot realize that NPF happened more than 2 months before in FT could have any significant impact on CCN concentrations and cloud properties in Dec/Jan. If at all, only detailed model simulation may give a robust answer concerning this conclusion.

**Reply from Author:** As stated above, we decide not to use model simulation because of

many unknown and uncertain parameters in the Antarctic free troposphere. The most important issue of this point is which aerosol particles derived from NPF in the Antarctic free troposphere and boundary layer can be survived to be grown to the critical diameter for CCN activation, or not. Furthermore, Williams et al. (2002) presented clearly that lifetime of aerosol particles with size of $D_p$ <60 nm in free troposphere was controlled by coagulation loss. In the revised manuscript, therefore, we focused on the aerosol lifetime by coagulation loss in troposphere. To our knowledge, number size distributions of aerosol particles with size with $D_p$ <100 nm are still unknown in the Antarctic free troposphere through the year. Therefore, we assumed that aerosol mixing ratios (i.e., number in respective size bins) in the free troposphere were as same as those at surface. Then, aerosol lifetime (e-folding time) was estimated using coagulation sink in each size. Vertical variations of the e-folding time showed clearly vertical gradient with longer lifetime at higher altitudes in all sizes ($D_p$ = 1 − 50 nm). The e-folding time at upper troposphere (ca. 8.3 km, 300 hPa) was 3 − 4 times longer than that at surface. Under the conditions with the assumption above (same mixing ratios), the e-folding time of aerosol particles larger than 20-30 nm exceed 30 days in middle free troposphere. Although size distributions of aerosol particles with size of $D_p$ <100 nm are still unknown in the Antarctic free troposphere, size distribution of aerosol particles with size larger than 300 nm was available over Syowa (1 − 2 order lower in free troposphere than at surface; Kizu et al., 2010).   If vertical gradient of aerosol number concentrations and size distributions of aerosol particles smaller than 100 nm was similar to that of aerosols with $D_p$ >300 nm under the background conditions, the longer e-folding time in the free troposphere is expected. Consequently, we conclude that the e-folding time of 1 − 2 months might be available for aerosol particles with size of $D_p$ >30 nm in the Antarctic free troposphere, with vertical gradient of the number concentrations of pre-existing particles. This discussion was added into the revised manuscript.

*Comment from Referee:* Chapters 3.4, line 293, Figures 8, and 9b: Does the calculated significance level refer to an ANOVA variance test, meaning that the corresponding distributions are significantly different on this level.

**Reply from Author:** ANOVA variance test was made for the revised manuscript. The result was added into the revised figure. Also, ANOVA test showed significant difference ($p$ = 0.0125).

*Comment from Referee:* Figure 5: Please specify the black lines and dots in the figure caption.

**Reply from Author:** Explanation of black lines and dots were added into the caption. The lines and dots showed the data numbers (n in y-axis).

All corrected parts and sentences were marked by red characters in the revised manuscript.

---

## Author Comment (AC2)

Reply to comments from Reviewer#2:

We would like to thank your helpful comments to improve our manuscript. All comments are responded and addressed in the current revise. Also, we added more careful analysis, for example aerosol lifetime and size distributions in air mass origins of BL and FT. Details are listed as follows.

*Comment from Referee:* The literature review is very poor, most of the references are not cited in the reference list, there is no mention to many Antarctic papers refering to other chemical component driving new particle formation (Sipila et al 2015, Jokinen et al., 2018), as many others discussing NPF in other stations (some of the Dr Weller studies are cited but much more recent work is not discussed).
**Reply from Author:** We added more explanation and comparison between our data and previous studies including recent studies as suggested from you.

*Comment from Referee:* Figure 10 seem to report a schematic presenting hypothesis. This seem to be related to the Ito et al., 1993 paper - but poor support is given by the data analyzed.
**Reply from Author:** Certainly, Ito (1993) provided schematic figure about likelihood of NPF in the Antarctic free troposphere. The most important issues on NPF in our hypothesis are NPF both in FT and BL, NPF enhanced by more UV radiation due to ozone hole, and their impact on aerosol population and clouds. These points were not included in Ito (1993). NPF in the Antarctic free troposphere was identified by previous works (Hara et al., 2011; Humphries et al., 2016; Lachlan-Cope et al., 2020). We added more careful analysis about aerosol lifetime in the Antarctic troposphere (section of 3.4 and Figure 8) in the revised manuscript. Vertical variations of aerosol lifetime (e-folding time) showed the e-folding time at upper troposphere (ca. 8.3 km, 300 hPa) was $3 - 4$ times longer than that at surface. Consequently, we conclude that the e-folding time of $1 - 2$ months might be available for aerosol particles with size of $D_p$ >20-30 nm in the Antarctic free troposphere. The longer lifetime is realistic range enough to lead to relation among NPF in the periods with ozone hole, aerosol population and CCN ability in the summer.

*Comment from Referee:* line 12 put year of measurements
**Reply from Author:** The year was added in the revised manuscript.

*Comment from Referee:* line 16 put some %, at the moment paper is Very desciprive

**Reply from Author:** Abundance (percentage) was added in the revised manuscript.

*Comment from Referee:* line 24 not sure it is seen in the data

**Reply from Author:** Vertical profiles of UV amounts have never been measured in the Antarctic troposphere to our knowledge. However, vertical variations of light intensity including UV show greater intensity at higher altitudes because of atmospheric scattering and absorption. This is basic knowledge in textbook.

*Comment from Referee:* line 40-50 poor literature review and most of the papers are not in the ref list

**Reply from Author:** Recent works and more explanation including review about NPF and condensable vapors in the Antarctic troposphere were added in the revised manuscript.

*Comment from Referee:* line 55-70 other chemical components are mentioned and found in other papers (amines, iodine, organics, ect)

**Reply from Author:** More explanation (review) about chemistry of condensable vapors (e.g., amines, iodine species, $NH_3$) in the Antarctic troposphere were added in the revised manuscript.

*Comment from Referee:* line 90-120 check equations with previous papers (Delmaso et al, and others)

**Reply from Author:** Dal Maso et al. (2002) was cited in the revised manuscript. Procedures to calculate coagulation sink were changed in the revised manuscript in accordance with Dal Maso et al. (2002).

*Comment from Referee:* line why 500 metres above ground? it is suggested to make a whole new analysis and a new calculation with multi ending point at ground level

**Reply from Author:** Considering uncertainty of the trajectory analysis in troposphere and high uncertainty at lower altitudes, we set initial height of 500m corresponding to upper boundary layer over Syowa, which were observed by the tethered balloon measurements in 2005 at Syowa Station (Hara et al., 2011a, 2013). We added some mention in the revised manuscript.

*Comment from Referee:* line 160 not sure if this is a valid classification, surely I am not familiar with these types of multifitting and not compared with other existing studies

(including polar aerosol size distributions and PMF and K means clustering).

**Reply from Author:** Multi-fitting by log-normal distribution is common way to compare aerosol size distributions. Some explanation and comparison with other works using multi-fitting in the Antarctic (e.g., Järvinen et al., 2013; Weller et al., 2015) were added into the revised manuscript.

*Comment from Referee:* line 155 not sure if the classification is valid. Additionally, the title reporte a full analysis but in essence this paper only talks about nucleation (and poorly analyzed)

**Reply from Author:** More explanation about modal distribution other than tri- and quad-modal structures were added into the revised manuscript.

*Comment from Referee:* line 150-160 not sure these are nucleation events, not clear from only one size distributions. How many full events, what types, duration, growth rate, examples, case studies).

**Reply from Author:** Because of lower aerosol number concentrations during winter – early spring, hourly-mean aerosol size distributions were obtained hardly through the year to identify NPF and to analyze growth rate after NPF. Thus, daily mean aerosol size distribution was analyzed and discussed in the present study. Details of respective NPF events during the periods with higher aerosol concentrations will be discussed elsewhere.

*Comment from Referee:* Again, there is not a % or any number, very qualitative

**Reply from Author:** Abundance (percentage) was added in the revised manuscript.

*Comment from Referee:* line 180-190 Blowing snow and sea spray, many studies are available in the literature in Antarctic regions and not described and compared here.

**Reply from Author:** We added more explanation and the specific references about sea-salt aerosols originated from the snow surface in the revised manuscript.

*Comment from Referee:* line 184. "and so on" seem a very poor way of reporting scientific pathways.

**Reply from Author:** We modified the sentence ("and so on" was removed in the revised manuscript).

*Comment from Referee:* line 210 the Na is not measured and not sure it is a valid method

to report. It is calculated and not sure valid.

**Reply from Author:** The concentrations of $Na^+$ were determined simultaneously with $SO_4^{2-}$ and $CH_3SO_3^-$ using ion chromatograph in this study (e.g., Hara et al., 2018). The procedures were listed in section of 2.1.

*Comment from Referee:* line 280-330 discussion and conclusions are more of a speculation section, not sure the data are showing this, and it is difficult to see SMPS data given the analysis carried out is poor.

**Reply from Author:** We believe that the most important issue of this point is which aerosol particles derived from NPF in the Antarctic free troposphere can be survived to be grown to the critical diameter for CCN activation, or not. Furthermore, Williams et al. (2002) presented clearly that lifetime of aerosol particles with size of $D_p$ <60 nm in free troposphere was controlled by coagulation loss. In the revised manuscript, therefore, we focused on the aerosol lifetime by coagulation loss in troposphere. To our knowledge, number size distributions of aerosol particles with size with $D_p$ <100 nm are still unknown in the Antarctic free troposphere through the year. Therefore, we assumed that aerosol mixing ratios (i.e., number in respective size bins) in the free troposphere were as same as those at surface. Then, aerosol lifetime (e-folding time) was estimated using coagulation sink in each size. Vertical variations of the e-folding time showed clearly vertical gradient with longer lifetime at higher altitudes in all sizes ($D_p = 1 - 50$ nm). The e-folding time at upper troposphere (ca. 8.3 km, 300 hPa) was $3 - 4$ times longer than that at surface. Under the conditions with the assumption above (same mixing ratios), the e-folding time of aerosol particles larger than 30 nm exceed 30 days in middle free troposphere. Although size distributions of aerosol particles with size of $D_p$ <100 nm are still unknown in the Antarctic free troposphere, size distribution of aerosol particles with size larger than 300 nm was available over Syowa ($1 - 2$ order lower in free troposphere than at surface; Kizu et al., 2010). If vertical gradient of aerosol number concentrations and size distributions of aerosol particles smaller than 100 nm was similar to that of aerosols with $D_p$ >300 nm under the background conditions, the longer e-folding time in the free troposphere is expected. Consequently, we conclude that the e-folding time of $1 - 2$ months might be available for aerosol particles with size of $D_p$ >30 nm in the Antarctic free troposphere, with vertical gradient of the number concentrations of pre-existing particles. This discussion was added into the revised manuscript.

*Comment from Referee:* line 525 figure 10 is very speculative and does not take into

account any paper published since the Ito et al 1993 paper. It also speculate that only FT NPF can make CCN whereas the BL NPF cannot make CCN.

**Reply from Author:** As stated above, vertical variations of e-folding time showed clear vertical gradient with longer lifetime at higher altitudes in all sizes ($D_p = 1 - 50$ nm). The e-folding time in the upper troposphere was $3 - 4$ times longer than that in boundary layer. The shorter lifetime in BL implies strongly that aerosol particles from NPF in BL were removed efficiently by coagulation except large growth rate and that aerosol particles from NPF were survived for longer time ($1 - 2$ months) in the free troposphere. This discussion was added into the revised manuscript. Knowledge obtained by other works since Ito (1993) was added into the schematic figure.

---

## Author Response (AR2)

Reply to reviewer's comments

We correct our manuscript based on reviewer's comments. Because authors pointed out some grammatical issues in the revised parts, the manuscript was checked carefully by us (authors) and native speaker.

**Reviewer' comment:** line 73: a reader not familiar with DMS chemistry may get the wrong impression that also SO2 is emitted by marine biogenic activity, which is not true as SO2 is the major reaction product of DMS in the atmosphere. Please modify this sentence to avoid misunderstandings.

**Reply form authors:** We agree with the comment from reviewer. We modified the sentence as follows.

Actually, $H_2SO_4$ in the Antarctic is converted dominantly via photochemical oxidation of dimethyl sulfide (DMS) released from biogenic activity in the ocean, and $SO_2$ derived from DMS oxidation (e.g., Minikin et al., 1998; Weller et al., 2015; Enami et al., 2017; Jang et al., 2019).

**Reviewer' comment:** lines 162-165: I am not convinced about the relevance of sea-salt emissions in this context. Although sea spray emissions extend to the ultrafine size range, can they really influence sub-20 nm sizes to interfere with determination of J5 using equations 5-7? The same issue is related to the statement on lines 330-331.

**Reply form authors:** As shown in previous work (Hara et al., 2011b), sea-salt particles with less volatility were distributed even in $D_p \leq 20$ nm during winter. Particularly, the number concentrations in ultrafine particles increased remarkably by strong emission of sea-salt aerosols from sea-ice areas under storm conditions during the winter. Details were discussed in Hara et al. (2011b). Because $J_5$ was estimated using the number of aerosol particles in size bins with range of $D_p = 5–20$ nm in this study, mixing of sea-salt particles during the winter can lead to the false values.

**Reviewer' comment:** line 173 and later: please be more specific in that Dpi is the number mean diameter of the mode (not e.g. mass mean diameter used by those people dealing with particle mass size distributions). Later on, the authors call this same diameter as "modal size", which is confusing. I would recommend keeping with the same term

throughout the paper. The text on lines 204-206 is particularly confusing: the authors should rather state that the number mean diameters of the mode(s) was(were) in the range(s) of xx-yy nm.

**Reply form authors:** We modified the sentence based on reviewer's comment, as follows.

In equation (9), $D_p$, $n$, $D_{p, i}$, $\sigma_i$, and $N_i$ respectively denote the particle diameter, mode number ($n$ = 1–4), modal size in mode i (i.e., mean diameter of the mode in aerosol number size distributions), modal standard deviation in mode $i$, and the aerosol number concentrations in mode $i$.

We keep to use "modal size" in the revised manuscript. Also, the descriptions were modified the sentences in Section 3.1 base on the later comment. The corrected sentences are written in lines of 208-211.

**Reviewer' comment:** lines 200-201: While the presence of quad-modal distributions is acceptable, I am not quite convinced about the reasoning here. Time-averaging data tends to smoothen details in it, so one would expect to fewer modes in daily-average distributions compared with shorter-average one. Unless the authors have a concrete evidence on their claim, I would recommend them to avoid statements like this, or at least say that this is only one possible explanation causing the difference between their and earlier studies.

**Reply form authors:** To avoid misunderstanding, the statement was removed in the revised manuscript.

**Reviewer' comment:** lines 213-214: the claim that particles grow to a few tens of nm immediately after NPF is very strange in this context. Considering the typical growth rates associated with NPF events reported in Antarctica, such growth will take a few hour in minimum, often a few days. "Immediately" is therefore not a proper word here.

**Reply form authors:** We modified the sentence based on reviewer's comment, as follows.

As demonstrated by Asmi et al. (2010), Kyrö et al. (2013), Järvinen et al. (2013), Weller et al. (2015), Jokinen et al. (2018), and Kim et al. (2019), aerosol particles were grown to a few tens of nanometers after NPF, even in the Antarctic troposphere during summer.

**Reviewer' comment:** line 229: lower solar radiation sounds like an understatement here. Should one rather say … in spite of almost total absence of solar radiation….?

**Reply form authors:** We modified the sentence as follows.

Surprisingly, tri-modal structures were identified even under dusk and polar night conditions during May–August.

**Reviewer' comment:** Section 3.2: When discussing different air masses, I would recommend keeping the word "air mass" or "air masses" everywhere in the text. Having just words like "MBL" or "continental FT" in the text may cause confusion, as such words usually refer to specific compartments in the atmosphere. For example saying that …structures were observed in MBL … (line 264) could be interpreted so that these structures were observed inside the MBL, although the authors mean that they were observed in the air mass type MBL.

**Reply form authors:** We agree with the comment from reviewer. We change the words like "air masses from continental FT" in the section and others.

**Reviewer' comment:** line 275: please specify what you exactly mean with the variability of NPF frequency. day-to-day, monthly or year-to-year variability?

**Reply form authors:** Here, we mean "monthly". We modified the sentence as follows.

The monthly occurrence (frequency) of the NPF, however, varied greatly at Syowa, Concordia and King Sejong.

**Reviewer' comment:** line 311-312: Do the authors mean …CN concentrations and their seasonal variation were…? "Features" is not a proper word here, and it is also unclear whether these features refer to CN or some other quantities discussed in the previous paragraph.

**Reply form authors:** We changed the sentence as follows.

CN concentrations and seasonal variations were similar to those measured at other

coastal stations (e.g., Weller et al., 2011; Fiebig et al., 2014).

**Reviewer' comment:** line 366-370: I do not fully agree with this reasoning. Transport of aerosol particle from the BL to the FT is usually rather inefficient, unless there is strong convective activity. Effective turbulent mixing mainly takes place within the BL (even though the BL may grow in height due to such mixing). Condensation sink of sea-salt particle is usually dominated by rather large sea-salt particles, and these are least effectively transported higher up in the atmosphere. In fact, the authors mention a large gradient between the BL and FT for >300 nm around their station (lines 413-415).

**Reply form authors:** Aerosol enhanced layer in BL and FT induced by rapid vertical mixing of aerosols (probably sea-salts) were observed over Syowa Station immediately after the storm conditions (Hara et al., 2014). This is the direct evidence from the observations. Additionally, sea-salt particles were distributed in ultrafine – coarse ranges during the winter and storm conditions as shown by Hara et al. (2011b). This vertical aerosol mixing is important and interesting in aerosol cycles in the Antarctic. It is true that we and others do not have knowledge of frequency of the vertical mixing events because aerosol observations in the FT under storm conditions have never been made. However, the strong and rapid vertical aerosol mixing can engender large influence on aerosol and atmospheric chemistry in FT. Therefore, the impact of vertical aerosol mixing is one of the future works.

**Reviewer' comment:** The figure numbering goes wrong after Figure 8. The real figure 9 is not referred to at all in the text, and Figures 10, 11 and 12 are referred to using wrong numbers. There are 2 figures with number S3.

**Reply form authors:** We check and correct the figure numbers in the text.

**Reviewer' comment:** The manuscript, especially the newly added text, contains many minor grammatical problems. One frequent problem are missing articles, especially in relation to the words "abundance" and "structure", but also elsewhere in the text. Please check out throughout the paper.

**Reply form authors:** We checked and corrected carefully typo and grammatical issues in the revised manuscript. Also, the revised manuscript was checked by native English speaker. Also, the grammatical things pointed by reviewer were modified in the revised

manuscript.